# Permissive lung neutrophils facilitate tuberculosis immunopathogenesis in male phagocyte NADPH oxidase-deficient mice

Eunsol Choi[1]☺, Hong-Hee Choi[1]☺, Kee Woong Kwon[1], Hagyu Kim[1], Ji-Hwan Ryu[2], Jung Joo Hong[3,4], Sung Jae Shin (ID)[1] *

1 Department of Microbiology, Institute for Immunology and Immunological Disease, Graduate School of Medical Science, Brain Korea 21 Project, Yonsei University College of Medicine, Seoul, South Korea,
2 Department of Biomedical Sciences, Graduate School of Medical Science, Brain Korea 21 Project, Yonsei University College of Medicine, Seoul, Korea, 3 National Primate Research Center, Korea Research Institute of Bioscience and Biotechnology, Cheongju, South Korea, 4 KRIBB School of Bioscience, Korea University of Science & Technology (UST), Daejeon, South Korea

☺ These authors contributed equally to this work.
* sjshin@yuhs.ac

**Data Availability Statement:** All relevant data are presented in the paper and supporting information.

## Abstract

NADPH oxidase 2 (NOX2) is an enzyme responsible for generating reactive oxygen species, primarily found in phagocytes. Chronic Granulomatous Disease (CGD), along with bacterial infections such as *Mycobacterium tuberculosis* (Mtb), is a representative NOX2-deficient X-linked disease characterized by uncontrolled inflammation. However, the precise roles of host-derived factors that induce infection-mediated hyperinflammation in NOX2-deficient condition remain incompletely understood. To address this, we compared Mtb-induced pathogenesis in *Nox2*[-/-] and wild type (WT) mice in a sex-dependent manner. Among age- and sex-matched mice subjected to Mtb infection, male *Nox2*[-/-] mice exhibited a notable increase in bacterial burden and lung inflammation. This was characterized by significantly elevated pro-inflammatory cytokines such as G-CSF, TNF-α, IL-1α, IL-1β, and IL-6, excessive neutrophil infiltration, and reduced pulmonary lymphocyte levels as tuberculosis (TB) progressed. Notably, lungs of male *Nox2*[-/-] mice were predominantly populated with CD11b[int]Ly6G[int]CXCR2[lo]CD62L[lo] immature neutrophils which featured mycobacterial permissiveness. By diminishing total lung neutrophils or reducing immature neutrophils, TB immunopathogenesis was notably abrogated in male *Nox2*[-/-] mice. Ultimately, we identified G-CSF as the pivotal trigger that exacerbates the generation of immature permissive neutrophils, leading to TB immunopathogenesis in male *Nox2*[-/-] mice. In contrast, neutralizing IL-1α and IL-1β, which are previously known factors responsible for TB pathogenesis in *Nox2*[-/-] mice, aggravated TB immunopathogenesis. Our study revealed that G-CSF-driven immature and permissive pulmonary neutrophils are the primary cause of TB immunopathogenesis and lung hyperinflammation in male *Nox2*[-/-] mice. This highlights the importance of quantitative and qualitative control of pulmonary neutrophils to alleviate TB progression in a phagocyte oxidase-deficient condition.

**Funding:** This work was supported by the National Research Foundation of Korea (NRF) grant, funded by the Ministry of Science and Information and Communication Technology (MSIT) under grant number RS-2023-00208115 to SJS and the Korea Research Institute of Bioscience and Biotechnology (KRIBB) Research Initiative Program under grant number KGM4572431 to SJS. The funders had no role in the study design, data collection and analysis, decision to publish, or preparation of the manuscript.

**Competing interests:** The authors have declared that no competing interests exist.

## Author summary

While tuberculosis (TB) remains a global threat to public health, the immunologic factors contributing to TB susceptibility are not yet fully defined. To enhance our understanding of TB immunopathogenesis, we utilized TB-susceptible male *Nox2-/-* mice to dissect the immunologic factors accelerating TB pathogenesis. In this study, we observed that male *Nox2-/-* mice infected with *Mycobacterium tuberculosis* (Mtb) exhibited increased lung hyperinflammation and mycobacterial burden compared to female *Nox2-/-* mice and WT mice. Exacerbated TB immunopathogenesis in male *Nox2-/-* mice was strongly correlated with an extreme infiltration of pulmonary neutrophils and the loss of pulmonary lymphocytes. The pulmonary neutrophils in male *Nox2-/-* mice displayed immature phenotypes and mycobacterial permissiveness. The depletion of total neutrophils or the AM80-induced maturation of neutrophils ameliorated TB pathogenesis in *Nox2-/-* mice, reducing pulmonary neutrophil counts, lung hyperinflammation, and mycobacterial load. Furthermore, neutralization of G-CSF mitigated TB pathogenesis in male *Nox2-/-* mice by decreasing immature neutrophil counts. Thus, we identified that G-CSF-mediated generation of permissive immature neutrophils contributes to TB susceptibility in male *Nox2-/-* mice.

## Introduction

*Mycobacterium tuberculosis* (Mtb) is the causative agent of Tuberculosis (TB), which remains a significant public health concern worldwide. Approximately, one-quarter of the world's population is estimated to have latent Mtb infection, with 5–10% of these individuals at risk of developing active TB at some point during their lifetime [1]. The factors that influence susceptibility to Mtb are still not completely understood, which hampers the control of this harmful pathogen [2–4]. There is extensive documentation indicating that males are more susceptible to TB than females, and this sexual bias is observed in various animal models including mice and humans [5,6]. The increased TB susceptibility in males is likely due to differences in immune cell functions as well as in their compositions [7,8]. Such host factor-mediated alterations in immune function may directly influence immune responses to Mtb infection. Considering the importance of immunological balance in maximizing anti-TB immunity, both excessive immune suppression or activation might be harmful to the host during TB progression [9–11]. These clues have led to the hypothesis that several host factors, such as biological sex and genetic distributions, which modulate inflammatory responses, may influence TB immunopathogenesis.

The phagocyte NADPH oxidase (NOX2, gp91$^{phox}$) is located in the lumen of phagosomes. Its primary function is to produce reactive oxygen species (ROS), which play a crucial role in protecting the host against a wide range of pathogens [12,13]. NOX2 also plays a vital role in regulating autophagy and cytokine/chemokine signaling [14–16]. The NOX2 complex is a multimeric enzyme composed of a single catalytic transmembrane heterodimer (gp91$^{phox}$ / p22$^{phox}$) and four cytosolic subunits (p40$^{phox}$ / p47$^{phox}$ / p67$^{phox}$ / Rac). Among the six major components of NOX2 complex, gp91$^{phox}$ acts as the catalytic core for ROS generation [17,18]. Individuals with impaired NOX2 function may develop chronic granulomatous disease (CGD), characterized by an increased susceptibility to fungal and bacterial infections and impaired inflammation control [19–21]. Since *Nox2* is located at Xp21.1 of the X chromosome, CGD caused by *Nox2* deficiency is inherited in an X-linked recessive manner, primarily

affecting young males [22,23]. Furthermore, individuals with X-linked CGD have a heightened vulnerability to mycobacterial infections [24]. This poses significant challenges for TB therapy and prevention in X-linked CGD patients as BCG vaccination cannot be implemented due to the high risks of BCGosis, which is a disseminated mycobacterial infection that can occur following BCG vaccination [25–27].

Although phagocyte NADPH oxidase-originated ROS play a crucial role in clearing intracellular pathogens, studies have shown that phagocyte NADPH oxidase deficiencies did not significantly alter mycobacterial growth compared to wild type (WT) animals. According to Cooper et al., p47$^{phox}$-deficient mice featured increase in lung mycobacterial load from 15 to 30 days post-infection. However, the increase in mycobacterial load was not sustained after 30 days post-infection [28]. Additionally, Olive et al. reported that male and female gp91$^{phox}$-deficient mice did not show an increase in lung mycobacterial load until 60 days or 20 weeks post-infection [29,30]. These researches give clues that NOX2 deficiency may modulate immune responses against Mtb, independently of ROS-mediated bactericidal effects. Recent studies support this idea as Thomas et al. demonstrated that gp91$^{phox}$-deficient mice displayed enhanced inflammation and lung neutrophil infiltration after Mtb challenge, which were mediated by IL-1 [12,31].

Dysregulated inflammatory responses and aberrant granuloma formation are hallmark features of X-linked CGD. X-linked CGD patients and NOX2-deficient mice exhibit uncontrolled formation of granulomas, characterized by an excessive influx of phagocytic myeloid cells to the site of infection [32,33]. NOX2-deficient phagocytes, such as macrophages and neutrophils, are major producers of pro-inflammatory cytokines [34,35], which can cause tissue damage [36,37]. Given that NOX2 deficiency-induced X-linked CGD can cause TB susceptibility along with lung hyperinflammation, we hypothesized that Mtb infection would cause uncontrolled inflammation and disruption of immune tolerance in NOX2-deficient mice. As the maintenance of immune tolerance is critical for effective anti-TB immunity, we further hypothesized that controlling the specific immunologic factor which causes Mtb infection-driven hyperinflammation would be crucial for ameliorating TB pathogenesis.

In this study, we established four experimental groups to investigate how two independent host factors (NOX2 deficiency and male sex) can modify the immunological responses against Mtb infection. As both male sex and NOX2 deficiency are risk factors for TB susceptibility, we hypothesized that male *Nox2*⁻/⁻ mice would exhibit severer TB progression than female *Nox2*⁻/⁻ mice and male WT mice. Additionally, we anticipated that specific immune factors would contribute to TB pathogenesis and hyperinflammation in male *Nox2*⁻/⁻ mice, which show a clear correlation to disease progression. Finally, we aimed to discover novel interventions that can effectively regulate these disease-promoting factors in male *Nox2*⁻/⁻ mice.

We found that male *Nox2*⁻/⁻ mice exhibited higher levels of lung inflammation and mycobacterial burden compared to female *Nox2*⁻/⁻ mice and WT mice following Mtb challenge. Male *Nox2*⁻/⁻ mice also featured elevated production of pro-inflammatory cytokines in the lungs, along with excessive infiltration of immature neutrophils and reduction of lymphocytes at infection sites. We also dissected the properties of aberrant permissive immature neutrophils, which were distinct from mature neutrophils and constituted the majority of immune cell populations in male *Nox2*⁻/⁻ mice. By reducing the number of total lung neutrophils or immature neutrophils, we could successfully mitigate TB pathogenesis in male *Nox2*⁻/⁻ mice. Finally, we determined that G-CSF, rather than other cytokines, is the dominant controller of immature neutrophil generation through *in vivo* neutralization of each cytokine. Collectively, we identified G-CSF driven permissive neutrophils as the key immunologic factor that facilitates TB immunopathogenesis and lung hyperinflammation in male *Nox2*⁻/⁻ mice.

Furthermore, we suggested novel therapeutic interventions targeting immature neutrophils, which might be potential for controlling TB in NOX2-deficient condition.

## Results

### Male *Nox2*[-/-] mice feature increased mycobacterial burden and lung hyperinflammation following aerosol infection with the Mtb K strain

Female WT, male WT, female *Nox2*[-/-], and male *Nox2*[-/-] mice were aerosol-infected with the Mtb K. At four weeks post-infection, the mice were autopsied to assess the severity of TB and immune function (**Fig 1A**). The mice were euthanized at the four weeks post-infection time point, when the increases in lung bacterial load (**S1A Fig**) and neutrophil infiltration (**S1B Fig**)—key pathological features of Mtb-infected *Nox2*[-/-] mice [12,31]—reached their peak. These measures began to increase from two weeks post infection, peaked at four weeks post-infection, and subsequently decreased until twelve weeks post-infection in *Nox2*[-/-] mice. Male *Nox2*[-/-] mice showed the highest lung bacterial load, spleen bacterial load, and lung inflammation. Female *Nox2*[-/-] mice featured the second-highest severity, while male and female WT mouse groups were ranked as 3rd and 4th, respectively (**Fig 1B and 1C**). Pro-inflammatory cytokines including IFN-γ, IL-1α, IL-1β, IL-17A, IL-6, TNF-α, G-CSF, and BAFF were significantly upregulated in the lungs of male *Nox2*[-/-] mice (**Fig 1D**). Conversely, anti-inflammatory cytokines including IL-4, IL-5, and TGF-β were not upregulated in the lungs of male *Nox2*[-/-] mice, although IL-10 was significantly upregulated (**Fig 1E**). This indicates that NOX2 deficiency plays a significant role in accelerating TB immunopathogenesis, as both male and female *Nox2*[-/-] mice feature exacerbated TB progression. Furthermore, we also found that male sex additionally promotes TB pathogenesis, as indicated by severer TB progression in male *Nox2*[-/-] mice.

### Pulmonary infiltration of neutrophils and loss of B lymphocytes are correlated with TB pathogenesis of male *Nox2*[-/-] mice

To determine which immune cell population is associated with dysregulated pro-inflammatory cytokine production, live lung cell compositions of Mtb infected mice were examined according to the attached flow cytometry gating strategy (**S1C Fig**). Male *Nox2*[-/-] mice exhibited the highest numbers and percentages of CD11b+Ly6G+ lung neutrophils, while female WT mice featured the lowest among the four groups (**Fig 2A**). On the other hand, male *Nox2*[-/-] mice showed a significant reduction in pulmonary CD90.2+ T cell percentages (**Fig 2B**), CD19+B220+ B cell counts (**Fig 2C**), and CD19+B220+GL7+FAS+ Germinal Center B cell counts (**Fig 2D**).

The disparity in TB progression peaked at four weeks post-infection, as male *Nox2*[-/-] mice exhibited significantly increased lung bacterial loads and pulmonary neutrophil counts at this time, compared to male WT mice and female *Nox2*[-/-] mice. The increase in lung CFU (Colony-forming unit) s and pulmonary neutrophils in male *Nox2*[-/-] mice began between two and three weeks post-infection and reached its maximum at four weeks post-infection. Interestingly, the disparity in pulmonary neutrophil counts between male *Nox2*[-/-] mice and other groups was more pronounced than the differences in lung CFUs at two and three weeks post-infection, as the latter did not show significant statistical differences, suggesting that the influx of pulmonary neutrophils occurs earlier than the increase in lung bacterial load (**Fig 2E**). T cell responses against mycobacterial antigen stimulations were also examined, as IFN-γ producing CD4+ T cells take crucial roles in anti-TB immunity [38,39]. Although male *Nox2*[-/-] mice featured the most exacerbated TB progression among the four groups, T cell responses against

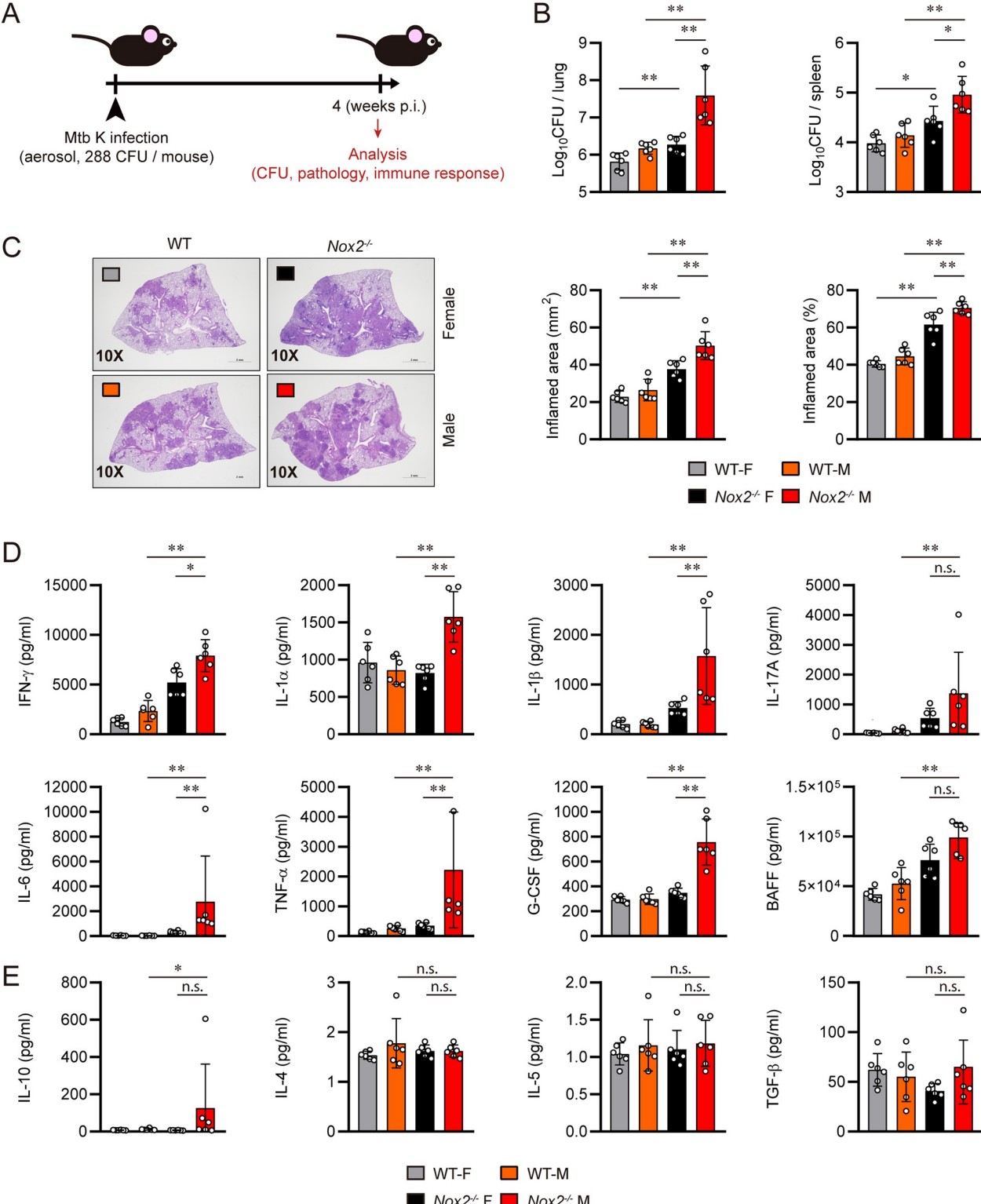

**Fig 1. Male *Nox2⁻/⁻* mice featured severe lung hyperinflammation and increased bacterial load upon Mtb infection. (A)** Experimental design for *in vivo* analysis of TB susceptibility. Six to seven-week-old female WT, male WT, female *Nox2⁻/⁻*, and male *Nox2⁻/⁻* mice (n = 6 per group) were aerosol-infected with Mtb K strain. At four weeks post-infection, all mice were autopsied, and immunological analysis, bacterial counting, and histopathological analysis were conducted (indicated by red arrow). Initial CFU = 288. **(B)** Mycobacterial CFUs in the lungs and spleens of each group at four weeks post-infection were analyzed by calculating the number of colonies and presented in bar graphs. **(C)** H&E staining was

performed on the superior lobes of the right lung at four weeks post-infection to visualize the gross lung pathology. The inflamed area of the H&E-stained samples was quantified in terms of percentage and square millimeters and presented in bar graphs. **(D)** IFN-γ, IL-1α, IL-1β, IL-17A, IL-6, TNF-α, G-CSF, and BAFF levels in Mtb-infected mouse lung lysates were measured by ELISA and LEGENDplex. **(E)** IL-10, IL-4, IL-5, and TGF-β levels in Mtb-infected mouse lung lysates were measured by ELISA and LEGENDplex. The cytokine levels are presented in bar graphs. The experiment was conducted twice. The data are presented as the mean ± SD of six mice in each group. The significance of differences was determined, using the One-way ANOVA test. *n.s.*, not significant. *$p < 0.05$, **$p < 0.01$.

Mtb antigens were strongly maintained (**S2 Fig**). The numbers of pulmonary CD11c[+]Siglec-F[+] alveolar macrophages, CD11b[+]CD64[+] macrophages, and CD11c[+]MHCII[+] dendritic cells were also significantly reduced in male *Nox2*[-/-] mice (**S3 Fig**). The immune cell dynamics in the lungs of mice infected with Mtb were visually represented using pie-charts and t-SNE plots. We observed that neutrophils, B cells, and T cells comprise the majority of pulmonary immune cells (**Fig 2E**). Furthermore, we found strong correlations between these immune cell populations and TB severity. The percentages of neutrophils displayed a strong positive correlation with TB severity, with high correlation coefficients and low p-values. Conversely, the percentages of T cells did not exhibit a significant correlation with TB severity, while the percentages of B cells demonstrated a reverse correlation with TB severity. The loss of pulmonary B cells also showed a significant negative correlation with neutrophil infiltration (**S4 Fig**). Thus, we hypothesized that TB susceptibility of male *Nox2*[-/-] mice is highly associated with the increase of pulmonary neutrophils.

## Pulmonary neutrophils of Mtb infected male *Nox2*[-/-] mice are predominantly composed of aberrant CD11b[int]Ly6G[int]CXCR2[lo]CD62L[lo] permissive immature neutrophils

It was not only the number of neutrophils that were altered in male *Nox2*[-/-] mice. Pulmonary neutrophils of male *Nox2*[-/-] mice featured CD11b[int]Ly6G[int] phenotypes, while neutrophils of other mouse groups featured CD11b[hi]Ly6G[hi] phenotypes. Atypical lung neutrophils of male *Nox2*[-/-] mice showed relatively lower expression of CD11b, Ly6G, Ly6C, CXCR2, and CD62L compared to WT neutrophils, along with higher expression of CXCR4 (**Fig 3A**). We categorized CD11b[+]Ly6G[+] pulmonary neutrophils into two distinct populations based on CXCR2 expression. According to previous studies, Ly6G[hi] or CXCR2[hi] lung neutrophils are categorized as mature neutrophils, whereas Ly6G[int] or CXCR2[lo] lung neutrophils are categorized as immature neutrophils [40–43]. We revealed that CD11b[+]Ly6G[+] pulmonary neutrophils of Mtb infected mice are consisted of CXCR2[hi] and CXCR2[lo] populations. Moreover, we figured out that CXCR2[hi] mature pulmonary neutrophils highly expressed CD62L, while CXCR2[lo] immature neutrophils featured low expression of CD62L through our own gating strategy (**Fig 3B**). The majority of *Nox2*[-/-] pulmonary neutrophils were CXCR2[lo]CD62L[lo] immature neutrophils, whereas WT pulmonary neutrophils comprised both immature and mature neutrophils. This suggests that the excessive accumulation of pulmonary neutrophils in male *Nox2*[-/-] mice is primarily mediated by immature neutrophil-specific lung infiltration (**Fig 3C**). We also analyzed neutrophil profiles from uninfected WT and *Nox2*[-/-] mice. The number of pulmonary total and immature neutrophils (**S5A and S5B Fig**), as well as bone marrow (BM) total and immature neutrophils (**S5C and S5D Fig**), were not increased in uninfected male *Nox2*[-/-] mice. Therefore, we concluded that the extreme pulmonary infiltration of CXCR2[lo]CD62L[lo] immature neutrophils is initiated by the pathogenic stimulus triggered by Mtb infection. CXCR2[lo]CD62L[lo] immature lung neutrophils displayed significantly lower expression levels of CD11b and Ly6G, which are major activation markers of neutrophils. They also featured lower FSC-A and SSC-A values compared to mature lung neutrophils, indicating that

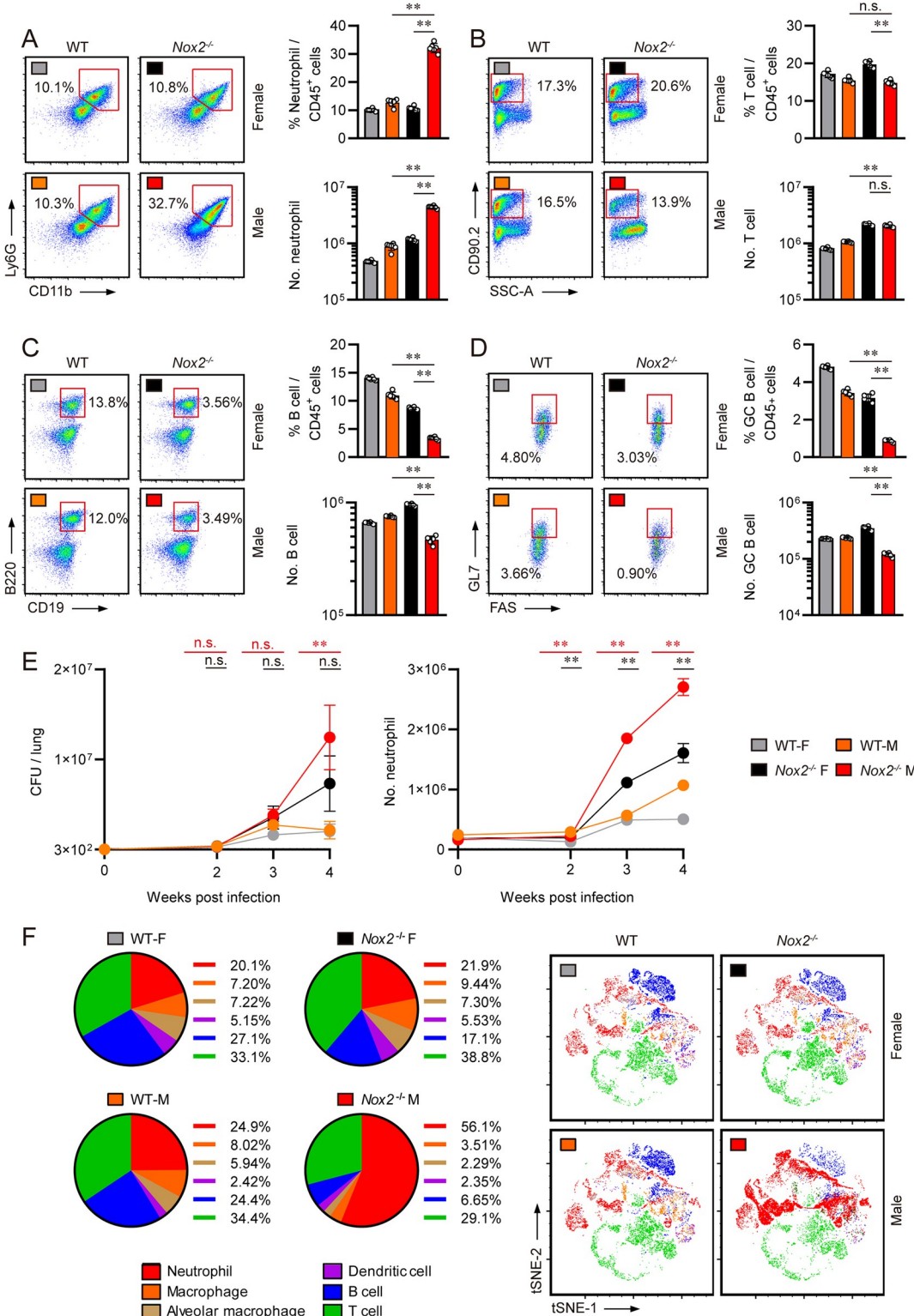

**Fig 2. Neutrophils are increased in the lungs of Mtb infected male *Nox2⁻/⁻* mice while B cells are diminished.** Pulmonary **(A)** CD11b⁺Ly6G⁺ neutrophil, **(B)** CD90.2⁺ T cell, **(C)** CD19⁺ B220⁺ B cell, and **(D)** CD19⁺ B220⁺ GL7⁺ FAS⁺ Germinal Center B cell populations of Mtb-infected mice at four weeks post-infection. The percentages of each immune cell among lung CD45⁺ cells and total cell counts are presented in bar graphs. **(E)** Pulmonary CFUs and neutrophil counts of female WT, male WT, female *Nox2⁻/⁻*, and male *Nox2⁻/⁻* mice were enumerated in a time-course dependent manner. Six-week old mice (n = 5

per group) were aerosol infected with Mtb K strain. The number of lung neutrophils were calculated at 0, 2, 3, and 4 weeks post-infection. Red bars and symbols represent statistical analysis between male *Nox2*[-/-] mice and male WT mice. Black bars and symbols represent statistical analysis between male *Nox2*[-/-] mice and female *Nox2*[-/-] mice. Initial CFU = 300. The experiment was conducted once. The data are presented as the mean ± SD of five or six mice in each group. The significance of differences was determined, using the One-way ANOVA test. *n.s.*, not significant. \*\**p* <0.01. **(F)** The compositions of the six major lung immune cell populations in Mtb-infected mice are presented in pie charts and t-SNE plots. The six major immune cells consist of neutrophils (red), macrophages (orange), alveolar macrophages (bright brown), dendritic cells (purple), B cells (blue), and T cells (green). The percentages of each immune cell among the sum of the six major immune cells are presented in pie charts.

CXCR2[lo]CD62L[lo] neutrophils are characterized by their smaller size and reduced granularity when compared to mature neutrophils (**Fig 3D**). To evaluate the permissiveness of pulmonary neutrophils, we isolated lung neutrophils from Mtb-infected mice and quantified intracellular bacterial counts. *Nox2*[-/-] pulmonary neutrophils, predominantly consisted of CXCR2[lo]CD62L[lo] immature neutrophils, exhibited higher mycobacterial permissiveness compared to WT pulmonary neutrophils (**Fig 3E**). To examine if CXCR2[lo]CD62L[lo] immature neutrophils are more permissive than CXCR2[hi]CD62L[hi] mature neutrophils, we infected male WT and *Nox2*[-/-] mice with YFP-expressing Mtb K strain. At four weeks post-infection, the mice were euthanized, and neutrophil permissiveness was evaluated using flow cytometry analysis (**S6A Fig**). We noted that neutrophils constituted the majority of YFP[+] cells in male *Nox2*[-/-] mice, and the number of Mtb-containing neutrophils were also increased. The mean fluorescence intensity (MFI) values of YFP in *Nox2*[-/-] lung neutrophils were substantially higher than those in WT mice, indicating increased mycobacterial permissiveness of *Nox2*[-/-] lung neutrophils (**S6B Fig**). Both immature and mature Mtb-containing neutrophil populations were enlarged in *Nox2*[-/-] mice. Notably, immature neutrophils displayed considerably higher YFP MFI values than mature neutrophils, suggesting their increased permissiveness to Mtb (**S6C Fig**). Conversely, the proportion of YFP[+] alveolar macrophages was lower in male *Nox2*[-/-] mice, and there was no significant difference in the number of YFP[+] alveolar macrophages between WT and *Nox2*[-/-] mice, although these cells exhibited relatively higher YFP MFI values compared to mature neutrophils (**S6D Fig**). The number of YFP[+] macrophages was elevated in *Nox2*[-/-] mice, but their YFP MFI values were relatively lower than those of immature neutrophils and alveolar macrophages (**S6E Fig**). In conclusion, we hypothesized that immature neutrophils are the primary permissive phagocytes in male *Nox2*[-/-] mice, considering both their dominance in population and their significant bacterial permissiveness as indicated by MFI values. These results suggest that permissive immature lung neutrophils may play a key role in driving TB pathogenesis and lung hyperinflammation in male *Nox2*[-/-] mice.

## Neutrophil depletion alleviates TB pathogenesis and lung hyperinflammation in male *Nox2*[-/-] mice

The exacerbated TB pathogenesis of male *Nox2*[-/-] mice was characterized by uncontrolled proinflammatory cytokine production and influx of permissive immature pulmonary neutrophils. Considering that neutrophils contribute to lung inflammation in *Nox2*[-/-] mice [20,44,45], and they are responsible for the increased mycobacterial burden in TB susceptible mouse models [46,47], we hypothesized that reducing Ly6G[+] neutrophils would alleviate TB pathogenesis in male *Nox2*[-/-] mice. Starting from two weeks post-infection, a Ly6G-specific monoclonal antibody (mAb) was intraperitoneally injected to WT and *Nox2*[-/-] mice to deplete neutrophils. At four weeks post-infection, the mice were autopsied (**Fig 4A**). Neutrophil depletion alleviated TB pathogenesis in both WT and *Nox2*[-/-] mice, by effectively reducing bacterial load in the lungs and spleens (**Fig 4B**) and lung inflammation (**Fig 4C**). IFN-γ, IL-1 α, IL-1 β, IL-6, TNF-

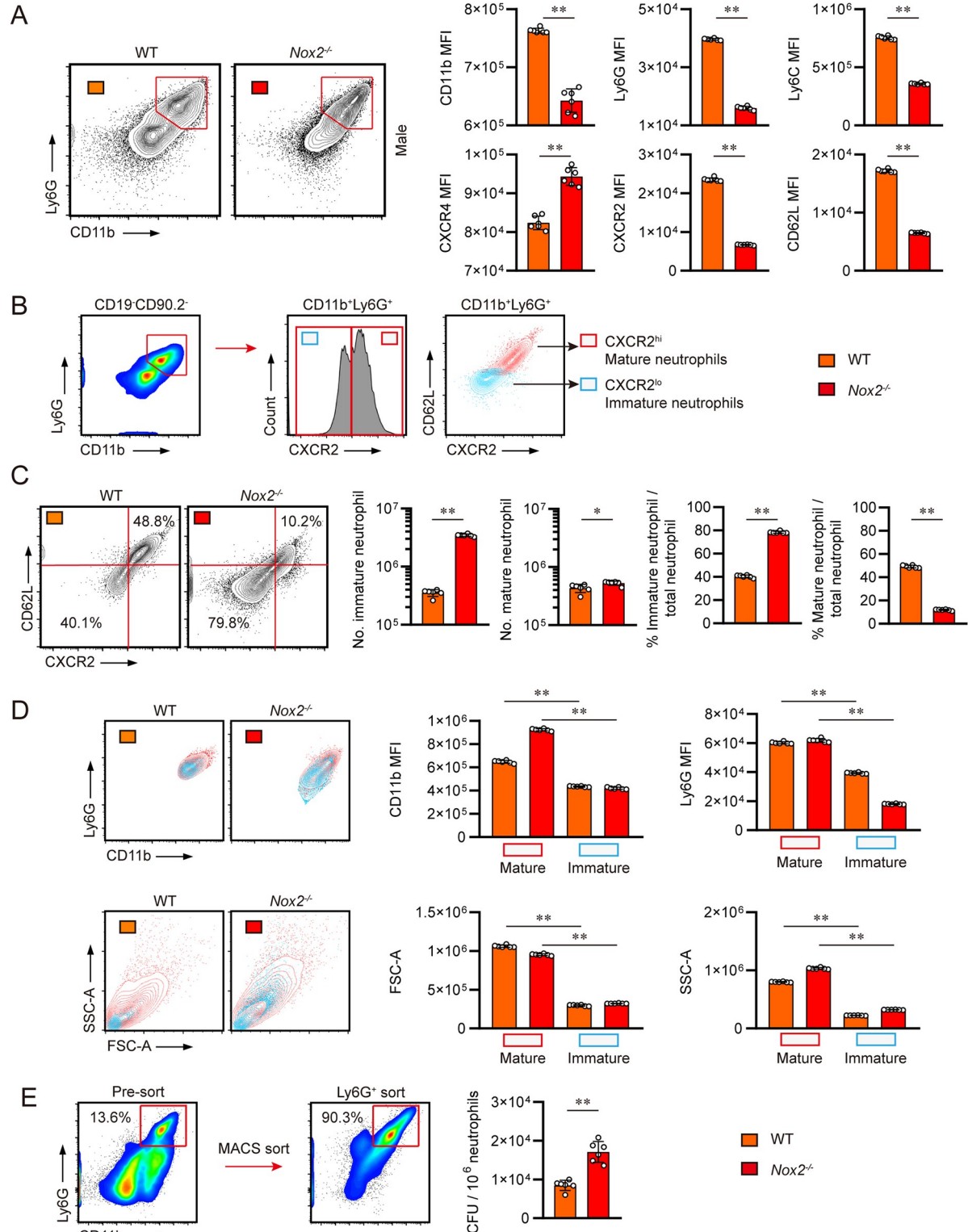

**Fig 3. CD11b^int^Ly6G^int^ CXCR2^lo^CD62L^lo^ immature neutrophils were dominant among the pulmonary neutrophils of Mtb infected male *Nox2⁻/⁻* mice. (A)** The expression levels of CD11b, Ly6G, Ly6C, CXCR4, CXCR2, and CD62L on CD11b⁺Ly6G⁺ neutrophils in the lungs of Mtb-infected mice at four weeks post-infection are presented. The mean fluorescent intensity (MFI) values of each molecule are presented in bar graphs. **(B)** The gating strategy used to distinguish immature and mature pulmonary neutrophils. CD11b⁺Ly6G⁺ neutrophils were classified into CXCR2^hi^ mature neutrophils (red) and CXCR2^lo^ immature neutrophils (bright blue). **(C)** CD62L expression

levels of CXCR2$^{hi}$ neutrophils and CXCR2$^{lo}$ neutrophils are presented into contour plots, demonstrating that mature neutrophils feature CXCR2$^{hi}$CD62L$^{hi}$ phenotypes while immature neutrophils feature CXCR2$^{lo}$CD62L$^{lo}$phenotypes. The numbers and percentages of immature and mature neutrophils among total pulmonary neutrophils in each group are presented in bar graphs, along with flow cytometry plots. **(D)** The expression levels of CD11b and Ly6G on mature and immature pulmonary neutrophils are presented in flow cytometry plots and bar graphs, along with FSC-A and SSC-A values of mature and immature neutrophils. **(E)** The purity of mouse lung neutrophils after Ly6G specific magnetic sorting and their bacterial permissiveness are presented in flow cytometry plots and bar graphs. Equal numbers of WT and *Nox2*$^{-/-}$ lung neutrophils were lysed and plated onto 7H10 agar plates to enumerate bacterial permissiveness. The data are presented as the mean ± SD of six mice in each group. The significance of differences was determined, using the One-way ANOVA test and Mann-Whitney U test. *$p < 0.05$, **$p < 0.01$.

α, and G-CSF levels were significantly downregulated in the lungs of male *Nox2*$^{-/-}$ mouse after neutrophil depletion (**Fig 4D**), while IL-10, IL-4, IL-5, and TGF-β levels were not significantly altered (**Fig 4E**). Therefore, we clarified that Ly6G$^+$ neutrophils are the actual cause of TB progression and lung hyperinflammation in male *Nox2*$^{-/-}$ mouse.

## Neutrophil depletion reduces immature lung neutrophils and restores pulmonary lymphocytes in Mtb infected male *Nox2*$^{-/-}$ mice

Ly6G$^+$ neutrophil depletion not only ameliorated TB pathogenesis, but also altered immune cell compositions in the lungs of male *Nox2*$^{-/-}$ mice. Neutrophil depletion removed most lung neutrophils in WT mice, and significantly reduced pulmonary neutrophil numbers in male *Nox2*$^{-/-}$ mice (**Fig 5A**). Surprisingly, it was highly effective at reducing CXCR2$^{lo}$CD62L$^{lo}$ pulmonary immature neutrophils in both WT and *Nox2*$^{-/-}$ mice (**Fig 5B**). Along with the reduction of immature pulmonary neutrophils, the number of B cells were increased in the lungs of Mtb infected male *Nox2*$^{-/-}$ mice (**Fig 5C**). However, neutrophil depletion did not significantly alter lung T cell population in male *Nox2*$^{-/-}$ mice (**Fig 5D**). This phenomenon was evident from the pie-charts and t-SNE plots illustrating lung immune cell dynamics, which clearly showed the reduction of pulmonary neutrophils and the restoration of B cells (**Fig 5E**). Consequently, we discovered that the reduction of pulmonary neutrophil counts can restore the balance between lung lymphocytes and neutrophils in male *Nox2*$^{-/-}$ mice.

## AM80 administration mitigates TB pathogenesis in male *Nox2*$^{-/-}$ mice and reduces pulmonary immature neutrophils

Among various strategies for the specific control of immature neutrophils, we considered RA treatment as a promising option. Previous studies have examined that retinoic acid (RA) or retinoic acid receptor (RAR) agonist administration can effectively promote the generation of fully differentiated mature neutrophils, which feature segmented nuclei and enhanced bactericidal capacity [48,49]. Moreover, Yamada et al. suggested the potential of RAs such as all-trans retinoic acid (ATRA) as a therapeutic agent against mycobacterial infection [50,51]. To investigate whether immature neutrophil-specific intervention can ameliorate TB pathogenesis in male *Nox2*$^{-/-}$ mouse, we utilized tamibarotene (AM80), a synthetic retinoid and a RAR agonist demonstrating 10 times greater potency than ATRA. AM80 induces the expression of key molecules, such as C/EBP, CD11b, and CXCR2 through the activation of RARα and RARβ [52], triggering a cascade of cellular pathways that generate fully functional mature neutrophils [53,54]. Starting from one-week post-infection, we orally administered 40 μg of AM80 per mouse, with a total of nine injections given over three weeks. At four weeks post-infection, the mice were autopsied (**Fig 6A**). AM80 administration significantly reduced mycobacterial load in the lungs and spleens (**Fig 6B**) and alleviated lung inflammation (**Fig 6C**) in male *Nox2*$^{-/-}$ mice. However, despite a reduction in pulmonary bacterial load, its effects were not prominent in reducing lung inflammation of male WT mice. AM80 administration downregulated

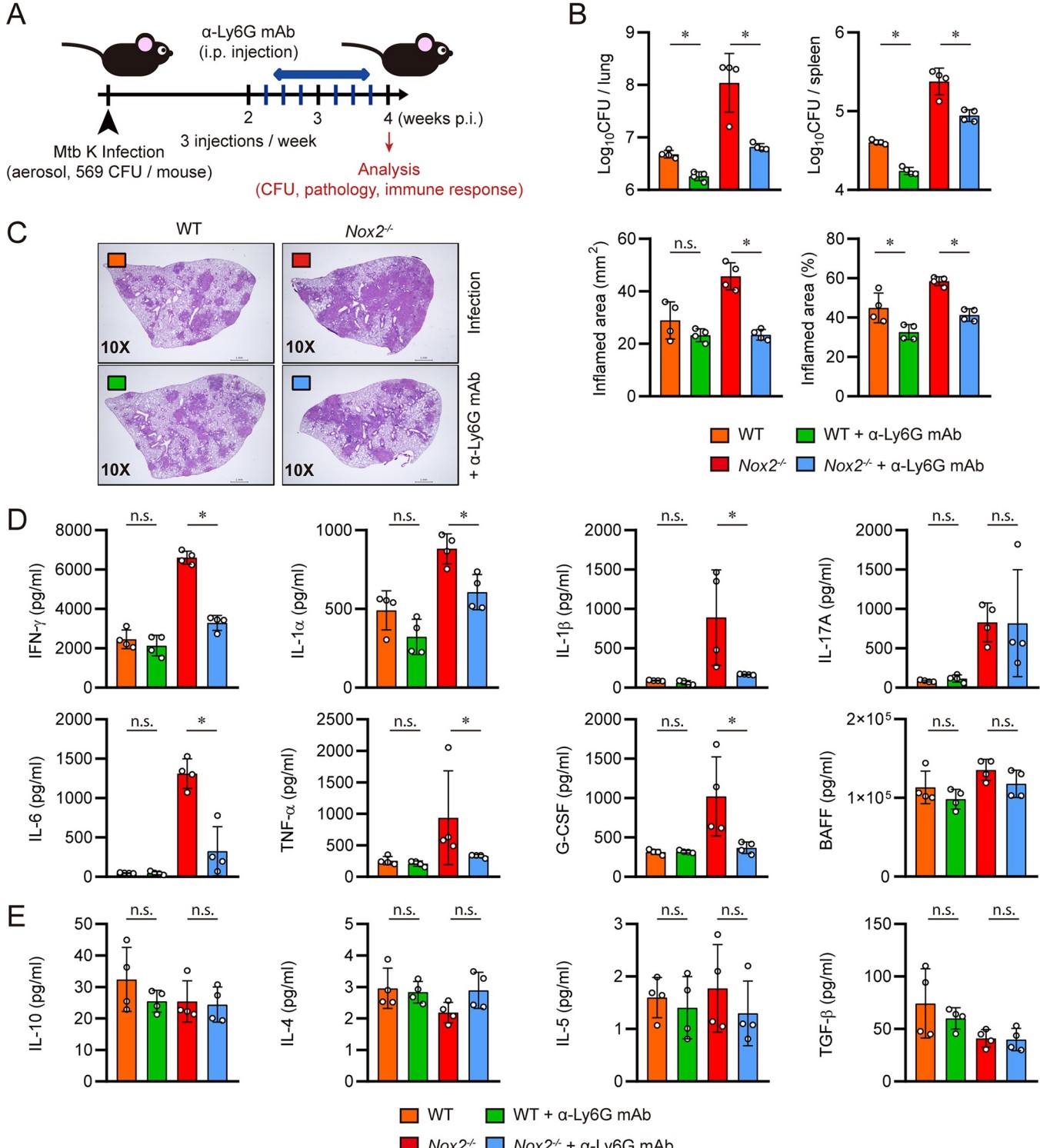

**Fig 4. Neutrophil depletion ameliorated TB pathogenesis in male *Nox2*$^{-/-}$ mice.** **(A)** Experimental design for *in vivo* depletion of neutrophils in Mtb infected mice. six-week old male WT and *Nox2*$^{-/-}$ mice (n = 4 per group) were aerosol infected with Mtb K strain. Starting from two weeks post-infection, 250 μg of anti-Ly6G mAb was intraperitoneally injected into each mouse three times a week (indicated by blue bars). At four weeks post-infection, all mice were autopsied, and immunological analysis, bacterial counting, and histopathological analysis were conducted (indicated by red arrow). Initial CFU = 569. **(B)** Mycobacterial CFUs in the lungs and spleens of each group at four weeks post-infection were analyzed by calculating the number of colonies and presented in bar graphs. **(C)** H&E staining was performed on the superior lobes of the right lung at four weeks post-infection to visualize the gross lung pathology. The inflamed area of the H&E-stained samples was quantified in terms of percentage and square millimeters and presented in bar graphs. **(D)** IFN-γ, IL-1α, IL-1β,

IL-17A, IL-6, TNF-α, G-CSF, and BAFF levels in Mtb-infected mouse lung lysates were measured by ELISA and LEGENDplex. (**E**) IL-10, IL-4, IL-5, and TGF-β levels in Mtb-infected mouse lung lysates were measured by ELISA and LEGENDplex. The cytokine levels are presented in bar graphs. The experiment was conducted twice. The data are presented as the mean ± SD of four mice in each group. The significance of differences was determined, using the One-way ANOVA and Mann-Whitney U test. *n.s.*, not significant. $^*p < 0.05$.

pulmonary expression of IFN-γ, IL-1 β, IL-17A, and G-CSF in male *Nox2⁻ᐟ⁻* mice. It did not substantially alter the expression of pro-inflammatory cytokines in male WT mice, although the expression of TNF-α was significantly reduced (**Fig 6D**). AM80 intervention did not alter the expression of IL-10, IL-4, IL-5, and TGF-β in both WT and *Nox2⁻ᐟ⁻* mice (**Fig 6E**). Total lung neutrophil counts (**Fig 6F**) and CXCR2^lo^CD62L^lo^ immature lung neutrophil counts (**Fig 6G**) were significantly decreased, and the portion of mature neutrophils among total lung neutrophils were increased in AM80-treated WT and *Nox2⁻ᐟ⁻* mice. While AM80 treatment effectively modified the accumulation of immature lung neutrophils, it did not exhibit significant effects in promoting Mtb-specific T cell responses in WT and *Nox2⁻ᐟ⁻* mice (**S7 Fig**). This suggests that the therapeutic potential of AM80 intervention in male *Nox2⁻ᐟ⁻* mice is largely contingent on the conversion of neutrophil maturity. To summarize, we specified immature lung neutrophils as the key subpopulation responsible for accelerating TB pathogenesis in male *Nox2⁻ᐟ⁻* mice.

## G-CSF predominantly regulates the generation of permissive immature pulmonary neutrophils in Mtb infected male *Nox2⁻ᐟ⁻* mice

To conclude the study, we aimed to identify specific immunologic factors responsible for the generation of CXCR2^lo^CD62L^lo^ permissive immature pulmonary neutrophils. Considering that pro-inflammatory cytokines upregulated in the lungs of Mtb-infected male *Nox2⁻ᐟ⁻* mice, such as IL-1, IL-6, and G-CSF accelerate the production of immature neutrophils through emergency granulopoiesis [55–57], we hypothesized that uncontrolled production of pro-inflammatory cytokines may trigger an explosive generation of immature neutrophils. To identify the causative cytokines, we administered mAbs to neutralize each cytokine in *Nox2⁻ᐟ⁻* mice. We confirmed that IFN-γ and IL-17A, key cytokines in anti-TB immunity for both innate and adaptive immune responses, are not associated with the generation of immature neutrophils. *In vivo* neutralization of IFN-γ or IL-17A increased mycobacterial burden, lung inflammation, pro-inflammatory cytokine levels, and pulmonary influx of immature neutrophils in female *Nox2⁻ᐟ⁻* mice (**S8 Fig**). Furthermore, *in vivo* neutralization of IL-6, IL-1α, and IL-1β also aggravated immature neutrophil-mediated TB pathogenesis and lung hyperinflammation in *Nox2⁻ᐟ⁻* mice, although IL-1 is reported to be crucial for mediating neutrophilic inflammation in NOX2-deficient conditions [12,36,37,44] (**S9 Fig**). Blockade of IL-1 receptor (IL-1R), which inhibits the function of both IL-1α, and IL-1β, exacerbated TB progression in *Nox2⁻ᐟ⁻* mice as well. We implemented two different *in vivo* blockade models: a three-week IL-1R blockade on female *Nox2⁻ᐟ⁻* mice and a two-week IL-1R blockade on male *Nox2⁻ᐟ⁻* mice, following the protocols from Olive et al. [12]. Both blockade models resulted in exacerbated TB progression, as evidenced by increased pulmonary bacterial load, lung inflammation, and neutrophil counts in both female and male *Nox2⁻ᐟ⁻* mice (**S10 Fig**). In contrast, *in vivo* neutralization of G-CSF, the main controller of granulopoiesis, remarkably ameliorated immature neutrophil-mediated TB pathogenesis in male *Nox2⁻ᐟ⁻* mice. Starting from one-week post-infection, G-CSF specific mAb was intraperitoneally administered, with a total of nine injections over a three-week period. At four weeks post-infection, the mice were autopsied (**Fig 7A**). While G-CSF neutralization did not alleviate TB progression in WT mice, it significantly reduced mycobacterial load in the lungs and spleens (**Fig 7B**) and mitigated lung

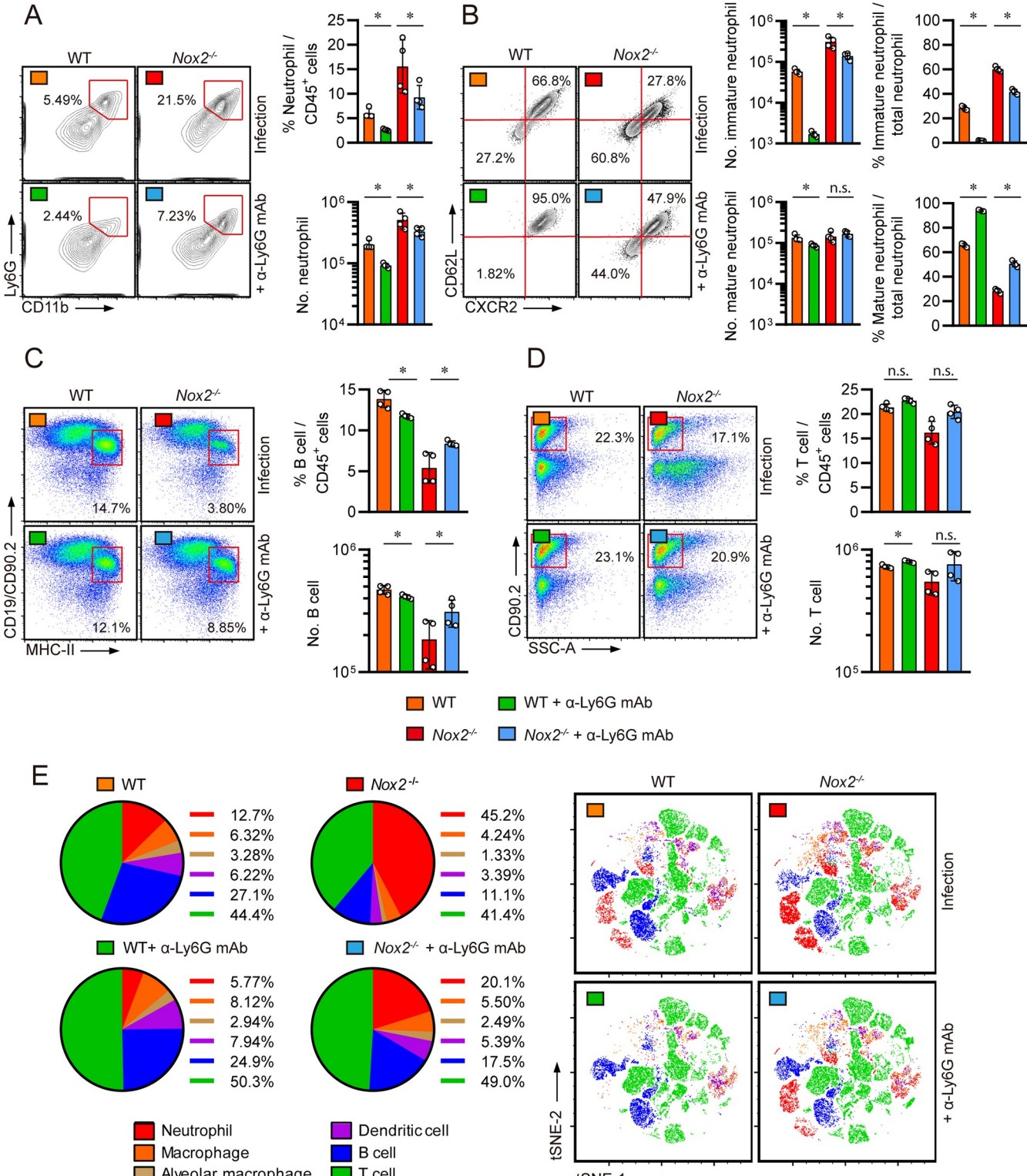

**Fig 5. Neutrophil depletion removed immature neutrophils and restored lymphocytes in the lungs of Mtb infected *Nox2*-/- mice.** Pulmonary **(A)** CD11b+Ly6G+ neutrophil, **(B)** CD11b+Ly6G+CXCR2loCD62Llo immature neutrophil, CD11b+Ly6G+CXCR2hiCD62Lhi mature neutrophil, **(C)** CD19+MHC-II+ B cell, and **(D)** CD90.2+ T cell populations of Mtb-infected mice at four weeks post-infection. The percentages of each immune cell among lung CD45+ cells or total neutrophils, and total cell counts are presented in bar graphs, along with flow cytometry plots. **(E)** The compositions of the six major lung immune cell populations in Mtb-infected mice are presented in pie charts and t-SNE plots. The six major immune cells consist of neutrophils (red),

macrophages (orange), alveolar macrophages (bright brown), dendritic cells (purple), B cells (blue), and T cells (green). The percentages of each immune cell among the sum of the six major immune cells are presented in pie charts. The data are presented as the mean ± SD of four mice in each group. The significance of differences was determined, using the One-way ANOVA and Mann-Whitney U test. *n.s.*, not significant. $^*p < 0.05$.

inflammation (**Fig 7C**) of male *Nox2⁻/⁻* mice. G-CSF neutralization also downregulated IFN-γ, IL-1α, IL-1β, IL-17A, IL-6, TNF-α, G-CSF, and BAFF levels specifically in the lungs of male *Nox2⁻/⁻* mice (**Fig 7D**), while IL-10. IL-4, IL-5, and TGF-β (**Fig 7E**) levels did not change. G-CSF neutralization decreased total lung neutrophil counts (**Fig 7F**) and CXCR2^lo^CD62L^lo^ immature lung neutrophil counts (**Fig 7G**) in male WT and *Nox2⁻/⁻* mice. The reduction of immature lung neutrophils was particularly pronounced and dramatically reflected in G-CSF neutralized male *Nox2⁻/⁻* mice. This underscores that the excessive G-CSF-mediated accumulation of pathogenic immature neutrophils is a specific event observed in TB susceptible male *Nox2⁻/⁻* mice Ultimately, we concluded that upregulated G-CSF is the immunologic trigger for the generation of permissive immature pulmonary neutrophils, which facilitates TB immunopathogenesis and lung hyperinflammation in male *Nox2⁻/⁻* mice (**Fig 8**).

## Discussion

Our study aimed to identify disease-promoting factors that are responsible for triggering Mtb-induced immunopathogenesis in the absence of phagocyte oxidase. We found that pulmonary infiltration of aberrant immature neutrophils is strongly associated with increased mycobacterial burden, lung hyperinflammation, and the loss of pulmonary lymphocytes in male *Nox2⁻/⁻* mice. We were able to remarkably alleviate TB progression in male *Nox2⁻/⁻* mice by reducing immature neutrophil counts. Overall, we identified G-CSF, rather than other host factors, as the initiator of immature pulmonary neutrophil-mediated TB immunopathogenesis in male *Nox2⁻/⁻* mice.

Progressed from previous studies on the roles of phagocyte oxidases in TB pathogenesis, our study proposed novel advocative findings. First, we have demonstrated that both sex difference and NOX2 deficiency can affect TB susceptibility in mice. In previous studies, gp91^phox^-deficient mice did not feature a significant increase in mycobacterial load when compared to WT mice [12,29–31]. Similarly, Duox1 or p47^phox^-deficient mice did not show an increase in lung mycobacterial load around four weeks post-infection [28,58]. Moreover, Olive et al. indicated that there was no variation in TB susceptibility between male and female gp91^phox^-deficient mice [12,31]. However, we displayed that male *Nox2⁻/⁻* mice exhibited a significantly higher lung mycobacterial load compared to WT mice and female *Nox2⁻/⁻* mice. This could be attributed to the use of the Mtb K strain, recognized for its heightened virulence in comparison to the H37Rv and H37Ra strains [59,60]. Additionally, as we focused *in vivo* analysis at the four weeks post-infection period, when TB severity in *Nox2⁻/⁻* mice is optimized, we were able to evaluate the exacerbated TB susceptibility of male *Nox2⁻/⁻* mice. This observation of time-point-dependent alterations in TB progression in *Nox2⁻/⁻* mice is particularly significant, as the pulmonary influx of neutrophils occurs before the explosive increase in mycobacterial CFUs. Such findings highlight the crucial role of constant pulmonary neutrophil influx, serving as a source of uncontrolled inflammation and a superior niche for Mtb, consequently driving TB pathogenesis. Our results newly indicate that biological sex is one of the crucial factors that accelerate TB pathogenesis in mice lacking phagocyte NADPH oxidase, which is inherited in an X-linked manner. Additionally, to the best of our knowledge, our study is the first to investigate the populations and functions of pulmonary lymphocytes in *Nox2⁻/⁻* mice during Mtb infection. We revealed that IFN-γ-producing CD4⁺ T cells were abundant, and antigen-specific Th1 responses were highly maintained in the lungs of male *Nox2⁻/⁻* mice.

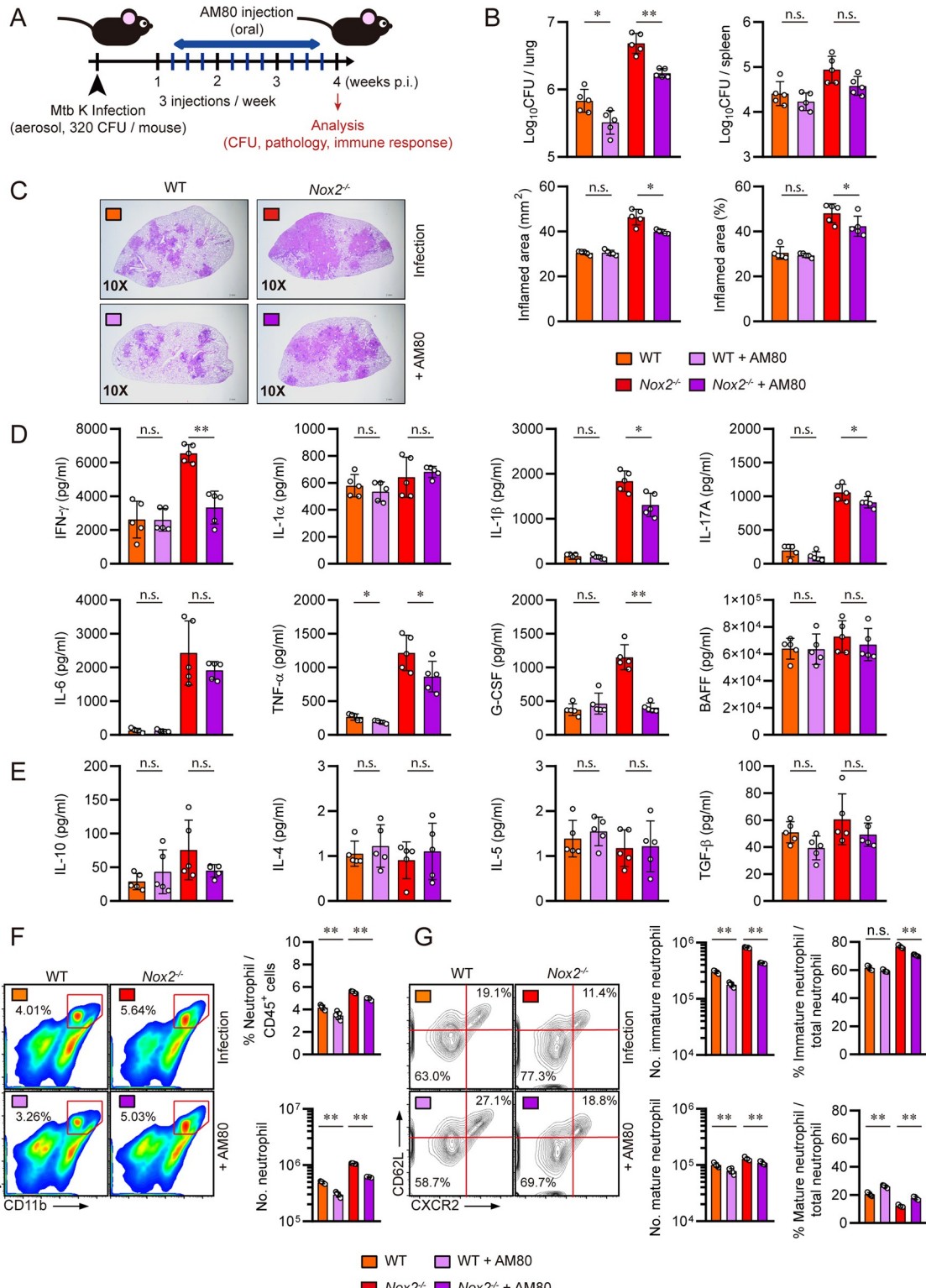

**Fig 6. AM80 administration mitigated TB pathogenesis in male *Nox2*-/- mice by reducing immature neutrophils. (A)** Experimental design for *in vivo* administration of AM80 in Mtb infected mice. Six-week old male WT and *Nox2*-/- mice (n = 5 per group) were aerosol infected with Mtb K strain. Starting from one week post-infection, 20 μg of AM80 was orally administered to each mouse three times a week (indicated by blue bars). At four weeks post-infection, all mice were autopsied, and immunological analysis, bacterial counting, and histopathological analysis were conducted (indicated by red arrow). Initial CFU = 320. **(B)**

Mycobacterial CFUs in the lungs and spleens of each group at four weeks post-infection were analyzed by calculating the number of colonies and presented in bar graphs. **(C)** H&E staining was performed on the superior lobes of the right lung at four weeks post-infection to visualize the gross lung pathology. The inflamed area of the H&E-stained samples was quantified in terms of percentage and square millimeters and presented in bar graphs. **(D)** IFN-γ, IL-1α, IL-1β, IL-17A, IL-6, TNF-α, G-CSF, and BAFF levels in Mtb-infected mouse lung lysates were measured by ELISA and LEGENDplex. **(E)** IL-10, IL-4, IL-5, and TGF-β levels in Mtb-infected mouse lung lysates were measured by ELISA and LEGENDplex. The cytokine levels are presented in bar graphs. Pulmonary **(F)** CD11b⁺Ly6G⁺ neutrophil, **(G)** CD11b⁺Ly6G⁺CXCR2^lo^CD62L^lo^ immature neutrophil, and CD11b⁺Ly6G⁺CXCR2^hi^CD62L^hi^ mature neutrophil populations of Mtb-infected mice at four weeks post-infection. The percentages of each immune cell among lung CD45⁺ cells or total neutrophils, and total cell counts are presented in bar graphs, along with flow cytometry plots. The experiment was conducted twice. The data are presented as the mean ± SD of five mice in each group. The significance of differences was determined, using the One-way ANOVA and Mann-Whitney U test. *n.s.*, not significant. $^*p < 0.05$. $^{**}p < 0.01$.

Hence, we concluded that the TB susceptibility of male *Nox2*[-/-] mice is not solely due to the inadequacy of Th1 responses, which contributes to TB susceptibility in various mouse models [61–63]. Lastly, we emphasized G-CSF as the primary inducer of immature neutrophil-mediated TB pathogenesis in male *Nox2*[-/-] mice, instead of other cytokines which remain essential for proper defense against Mtb infection. Chao et al. reported that the blockade of IL-1β or IL-1R can alleviate mycobacterial infection in phagocyte oxidase-deficient mice [12,37]. However, our data indicated that both IL-1α and IL-1β are required for proper defense against Mtb. IL-1α and IL-1β act as the upstream controllers of excessive G-CSF production in *Nox2*[-/-] mouse model [36,44], but they also play protective roles of in anti-TB immunity, such as activating myeloid cells in general, beyond just neutrophils [64–66]. We suggest that such protective roles of IL-1 signaling are especially crucial in the early phases of mycobacterial infection. By demonstrating that initiating IL-1R blockade at either one week or two weeks post-infection time point exacerbates TB progression in *Nox2*[-/-] mice, we highlighted the dominance of G-CSF over IL-1 in neutrophil-mediated TB pathogenesis.

Although *Nox2*[-/-] mice are prone to autoimmune diseases such as rheumatoid arthritis [67,68], Mtb-induced lung hyperinflammation in *Nox2*[-/-] mice was not primarily associated with impaired Th1 responses or autoreactive T cells. We observed that pulmonary neutrophils are the actual cause of lung hyperminflammation, and the loss of pulmonary B cells is more closely associated with neutrophilic inflammation rather than alterations in T cell responses. Our research may offer deeper insights into TB susceptible animal models that are not associated with altered T cell function. We explicitly characterized immature lung neutrophils and identified them as the primary drivers of TB immunopathology in male *Nox2*[-/-] mice. While various studies have emphasized the importance of neutrophils in promoting lung inflammation in *Nox2*[-/-] mice undergoing bacterial, parasitic, or fungal challenge [36,45,69–71], their properties were not dissected in detail. We uncovered that *Nox2*[-/-] immature pulmonary neutrophils feature CD11b^int^Ly6G^int^ and CXCR2^lo^CD62L^lo^ phenotypes, altered size and granularity, and increased mycobacterial permissiveness. Still, we need transcriptomic or metabolomic analysis of CD11b^int^Ly6G^int^CXCR2^lo^CD62L^lo^ immature neutrophils to elucidate the precise mechanisms underlying neutrophilic inflammation and mycobacterial permissiveness. We proposed novel interventions with therapeutic potentials to enable proper control of immature neutrophils. The effectiveness of neutrophil depletion in alleviating TB pathogenesis and Mtb-induced lung hyperinflammation was formerly demonstrated in TB susceptible *Nos2*[-/-] and *TLR2*[-/-] mice [72,73], as well as in WT mice suffering from chronic TB [74]. We firstly provided the therapeutic potential of neutrophil depletion in alleviating TB pathogenesis specifically in *Nox2*[-/-] mice. We also demonstrated the effectiveness of AM80 in alleviating TB pathogenesis. While Yamada et al. illustrated that ATRA intervention mitigated TB progression by increasing pulmonary T cell counts or reducing pulmonary myeloid-derived suppressor cells (MDSCs) [50,51,75–77], AM80 has not been tested on animal models as a therapeutic

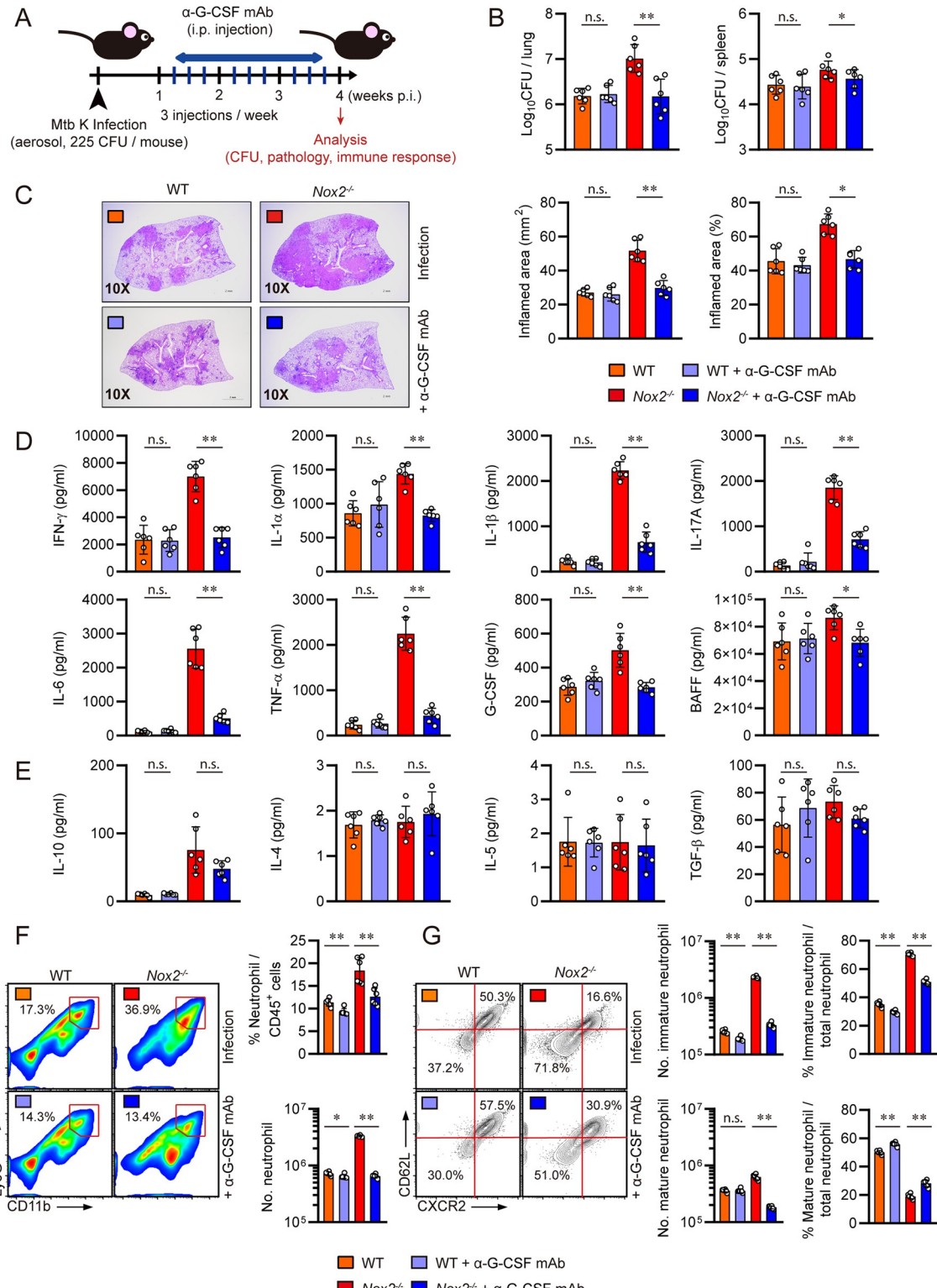

**Fig 7. G-CSF neutralization ameliorated TB pathogenesis of male *Nox2*-/- mice by suppressing generation of immature pulmonary neutrophils. (A)** Experimental design for *in vivo* neutralization of G-CSF in Mtb infected mice. Six-week old male WT and *Nox2*-/- mice (n = 6 per group) were aerosol infected with Mtb K strain. Starting from one week post-infection, 30 μg of anti-G-CSF mAb was intraperitoneally administered to each mouse three times a week (indicated by blue bars). At four weeks post-infection, all mice were autopsied, and immunological analysis, bacterial counting, and histopathological analysis were conducted

(indicated by red arrow). Initial CFU = 225. **(B)** Mycobacterial CFUs in the lungs and spleens of each group at four weeks post-infection were analyzed by calculating the number of colonies and presented in bar graphs. **(C)** H&E staining was performed on the superior lobes of the right lung at four weeks post-infection to visualize the gross lung pathology. The inflamed area of the H&E-stained samples was quantified in terms of percentage and square millimeters and presented in bar graphs. **(D)** IFN-γ, IL-1α, IL-1β, IL-17A, IL-6, TNF-α, G-CSF, and BAFF levels in Mtb-infected mouse lung lysates were measured by ELISA and LEGENDplex. **(E)** IL-10, IL-4, IL-5, and TGF-β levels in Mtb-infected mouse lung lysates were measured by ELISA and LEGENDplex. The cytokine levels are presented in bar graphs. Pulmonary **(F)** CD11b⁺Ly6G⁺ neutrophil, **(G)** CD11b⁺Ly6G⁺CXCR2ˡᵒCD62Lˡᵒ immature neutrophil, and CD11b⁺Ly6G⁺CXCR2ʰⁱCD62Lʰⁱ mature neutrophil populations of Mtb-infected mice at four weeks post-infection. The percentages of each immune cell among lung CD45⁺ cells or total neutrophils, and total cell counts are presented in bar graphs, along with flow cytometry plots. The experiment was conducted three times. The data are presented as the mean ± SD of six mice in each group. The significance of differences was determined, using the One-way ANOVA test. *n.s.*, not significant. *$p < 0.05$. **$p < 0.01$.

agent against TB. We highlighted the efficacy of AM80 in controlling TB pathogenesis in a susceptible mouse model, focusing on its role in reducing inflammatory neutrophils rather than altering T cell or macrophage responses. We have newly revealed that AM80 intervention does not alter Mtb-specific T cell responses in *Nox2*⁻/⁻ mice. Although AM80 may affect the phagocytic capacities of *Nox2*⁻/⁻ macrophages, we have excluded the effects on macrophages because AM80 did not alter the phagocytosis of *Nox2*⁻/⁻ BMDMs in our setup studies. Since AM80

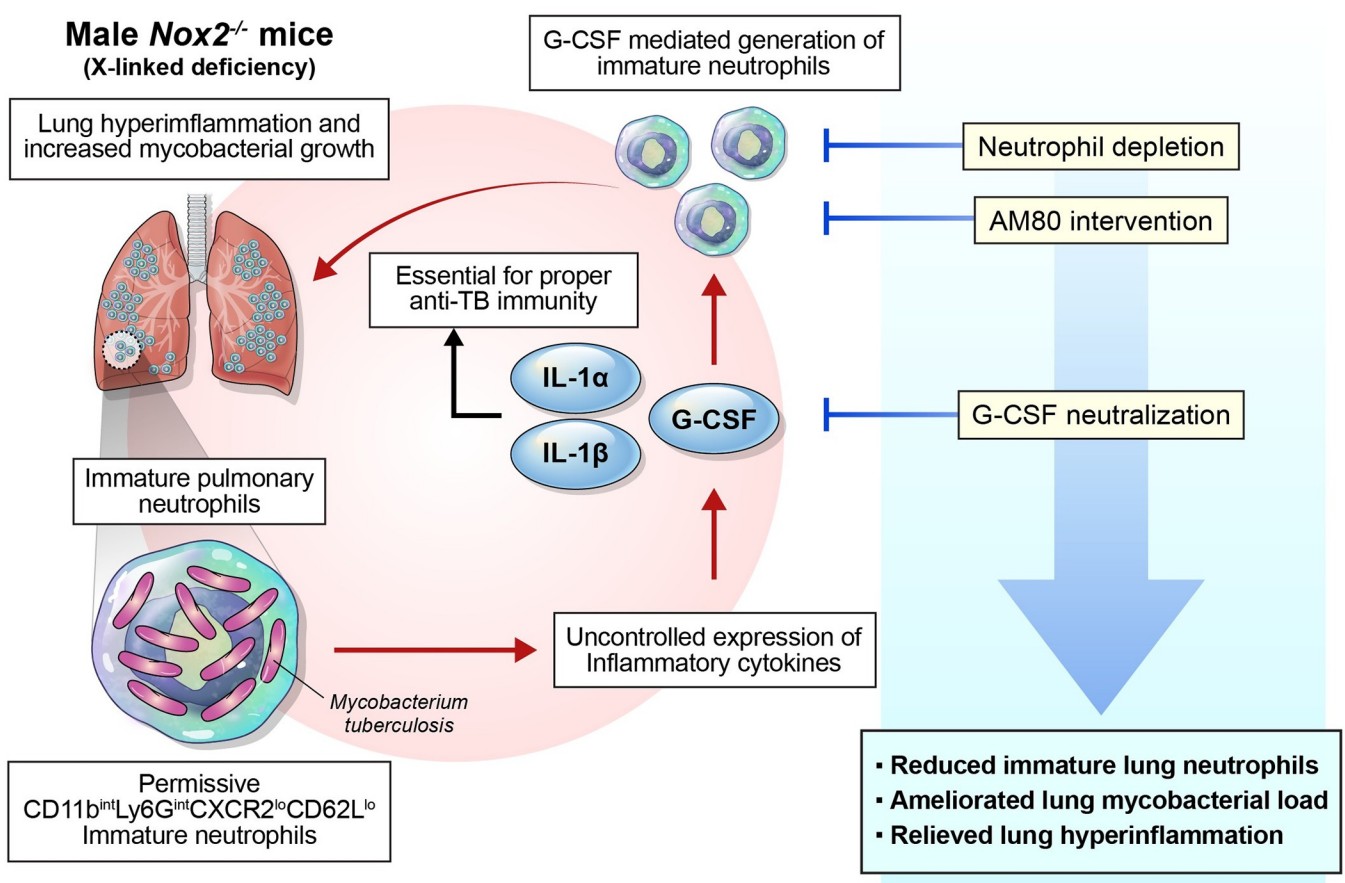

**Fig 8. The results of the study are summarized in a visual abstract.** CD11bⁱⁿᵗLy6GⁱⁿᵗCXCR2ˡᵒCD62Lˡᵒ permissive immature pulmonary neutrophils are responsible for lung hyperinflammation and increased mycobacterial load in male *Nox2*⁻/⁻ mice. Among the upregulated pro-inflammatory cytokines, G-CSF, rather than IL-1, plays a dominant role in generating immature neutrophils. The control of immature neutrophils through neutrophil depletion, AM80 intervention, and G-CSF neutralization has shown promising results in ameliorating TB immunopathogenesis in male *Nox2*⁻/⁻ mice.

shows enhanced potency compared to ATRA and is free from the side effects associated with RAR-γ activation [78,79], AM80 intervention may offer superior outcomes for establishing host-directed TB therapies in susceptible models compared to ATRA. Furthermore, our study newly discovered that G-CSF neutralization can alleviate TB pathogenesis in *Nox2*[-/-] mice. These attempts may provide insights into controlling TB progression in susceptible animal models with excessive pulmonary granulocyte influx.

Still, our study requires discussions on the following limitations. AM80 intervention was effective at mitigating TB pathogenesis in WT mice, while G-CSF neutralization did not clearly alter TB progression in this mouse model. This is considered to be originated from two different reasons. Firstly, unlike male *Nox2*[-/-] mouse model, WT mice do not show excessive and constant G-CSF production in response to Mtb infection. Therefore, Mtb infected WT mice may have an insufficient amount of circulating G-CSF which should be depleted through G-CSF neutralization. The second reason is that G-CSF and AM80 promotes myeloid cell generation via independent signaling pathways. AM80 induces neutrophil differentiation and maturation through the activation of RAR, while G-CSF is not involved in RAR signaling or C/EBPβ activation, rapidly generating immature neutrophils [52]. Therefore, we conclude that AM80 featured therapeutic capacity in both mouse strains, while G-CSF neutralization specifically showed potency in male *Nox2*[-/-] mouse model. While anti-Ly6G mAb injection successfully removed pulmonary neutrophils in WT mice, it failed to fully eliminate the cells in *Nox2*[-/-] mice. This outcome is thought to stem from the nature of the intraperitoneal injection of anti-Ly6G mAb (1A8). Boivin et al. noted that intraperitoneal injection of anti-GR1 or Ly6G monoclonal antibodies may not fully eliminate neutrophils, as newly synthesized young neutrophils can replenish the blood or organs even after depletion [80]. Furthermore, anti-Ly6G mAb mediates slow neutrophil depletion through Fc-dependent opsonization and phagocytosis of neutrophils by macrophages. Considering that NOX2 deficiency leads to both G-CSF mediated constant neutrophil production and macrophage malfunction, we conclude that these two factors contributed to the incomplete depletion of pulmonary neutrophils in male *Nox2*[-/-] mice. A similar explanation can be given for the persistence pro-inflammatory cytokines in the lungs, even after mAb-mediated cytokine neutralization. This inefficiency is likely due to the nature of intraperitoneal injections, which effectively target circulating cells or cytokines but may not completely reach lung-residing targets. Additionally, since *Nox2*[-/-] neutrophils are inflammatory, they are likely the source of the cytokines that were still detected after intraperitoneal depletion. To optimally deplete such targets, a combination of intraperitoneal and intratracheal injections is likely necessary to effectively target and deplete cytokines residing in the lung. The disparity in the effects of IL-1R blockade on *Nox2*[-/-] mice also warrants discussion. Despite following protocols from Olive et al. [12], initiating IL-1R blockade from two weeks post-infection exacerbated TB in male *Nox2*[-/-] mice. We conclude that differences in the infection model system contributed to this discrepancy. Specifically, our study differed from previous ones in terms of the mycobacterial strain virulence, initial infectious doses, and the ages of the mice used. IL-1R blockade may suppress neutrophil-mediated TB pathogenesis in a less severe infection model, but it was ineffective in our system, which involved a severe infection with a high dose of the highly virulent Mtb K strain.

Despite the abundance of T cell responses, male *Nox2*[-/-] mice displayed severe TB progression. Accumulating evidences suggest that disrupted T cell responses during Mtb infection may diminish protection against the infection. For instance, the deficiency of the T cell inhibitory receptor PD-1 leads to a significant enhancement of T cell responses and an increase in the pulmonary mycobacterial burden [81,82]. Given that neutrophil depletion and G-CSF neutralization decreased the pulmonary levels of IFN-γ and IL-17A in male *Nox2*[-/-] mice, maintaining the balance between immune activation and suppression might be more crucial for the

proper control of TB than the amount of cytokines produced. To gain a deeper understanding of TB pathogenesis, it is crucial to identify immunologic factors that disrupt the proper activation of Th1 responses. The absence of pulmonary B cells in the presence of BAFF, TNF-α, and IL-6, which are B cell activating cytokines, remains a question to be solved. As pulmonary B cells play crucial roles in suppressing neutrophil-mediated lung inflammation through direct interaction [83,84], their absence may contribute to TB susceptibility in *Nox2*<sup>-/-</sup> mice. Thus, elucidating the association between the increase in neutrophils and the decrease in B cells may contribute to a better understanding of anti-TB immunity. We investigated the impacts of AM80 intervention at four weeks post-infection. However, considering the broad roles of retinoic acids in modulating macrophage and T cell responses [50,51], understanding the significance of non-neutrophil factors affected by AM80, such as T cell responses, recruited macrophage functions, and alveolar macrophage functions at early stages of Mtb infection, can significantly enhance our understanding of the pathogenesis of TB in susceptible animal models.

Collectively, our study revealed that phagocyte NADPH oxidase deficiency contributes to exacerbated TB immunopathogenesis in a sex-dependent manner. This is attributed to the excessive pulmonary infiltration of immature neutrophils in the presence of sufficient Th1 responses. Therapeutic interventions targeting the generation of G-CSF driven immature pulmonary neutrophils successfully ameliorated TB immunopathogenesis in male *Nox2*<sup>-/-</sup> mice. Our study provides a comprehensive understanding of TB pathogenesis and highlights potential therapeutic targets for proper TB control in the absence of phagocyte-specific NADPH oxidase.

## Materials and methods

### Ethics statements and study approval

All animal experiments followed the regulations set by the Korean Food and Drug Administration. The Ethics Committee and Institutional Animal Care and Use Committee (2019–0174; C57BL/6N, *Nox2*<sup>-/-</sup>, 2022–0140; C57BL/6N, *Nox2*<sup>-/-</sup>) at Yonsei University Health System (Seoul, Korea) granted approval for each experimental protocol.

### Mice

Six- to seven-week-old female and male C57BL/6N mice were obtained from Japan SLC, Inc. (Shizuoka, Japan). Six- to seven-week-old female and male *Nox2*<sup>-/-</sup> mice were provided by Dr. Yun Soo Bae (Ewha Womans University, Seoul, South Korea). The mice were housed in a specific pathogen-free (SPF) environment with barriered conditions at the Yonsei University Medical Research Center SPF facility.

### Bacterial culture and infection protocols

Purchase of the Mtb K strain, mycobacterial culture, and aerosol infection processes were executed according to the referenced studies [85,86]. The mycobacteria were cultured in 7H9 broth supplemented with 10% oleic acid-albumin-dextrose-catalase (OADC; Difco Laboratories, MD) and 2 μg/ml of mycobactin J (Allied Monitor, Fayette, MO) for 4 weeks at 37˚C. Following cultivation, all bacteria were washed three times with 10 mM phosphate-buffered saline (PBS; pH 7.4). Bacterial cell pellets were collected after centrifugation, and small aliquots were stored at -80˚C until use. The Mtb K strain used for *in vivo* challenges was PDIM positive. YFP vectors for the generation of YFP⁺ Mtb K strain were kindly gifted by Dr. Christopher Sassetti. In order to deliver bacteria to the lungs of each mouse, all mice were exposed to at least 200 viable mycobacteria in an inhalation chamber for 60 minutes.

### *In vivo* treatments for neutrophil control

From two weeks post-infection, 250 µg/mouse of anti-Ly6G mAb (clone: 1A8, Bio X Cell, West Lebanon, NH, USA), 200 µg/mouse of anti-IL-1R mAb (clone: JAMA-147, Bio X Cell, West Lebanon, NH, USA), 200 µg/mouse of anti-IFN-γ mAb (clone: H22, Bio X Cell, West Lebanon, NH, USA), and 250 µg/mouse of anti-IL-17A mAb (clone: 17F3, Bio X Cell, West Lebanon, NH, USA) mAb were intraperitoneally administered. The antibodies were diluted in PBS and administered three times a week for a total duration of two weeks. From one-week post-infection, 30 µg/mouse of anti-G-CSF mAb (clone: #67604, R&D Systems, Minneapolis, MN, USA), 200 µg/mouse of anti-IL-1R mAb (clone: JAMA-147, Bio X Cell, West Lebanon, NH, USA), 400 µg/mouse of anti-IL-6 mAb (clone: MP5-20F3, Bio X Cell, West Lebanon, NH, USA), 200 µg/mouse of anti-IL-1α mAb (clone: ALF-161, Bio X Cell, West Lebanon, NH, USA), and 200 µg/mouse of anti-IL-1β mAb (clone: B122, Bio X Cell, West Lebanon, NH, USA) were intraperitoneally administered. The antibodies were diluted in PBS and administered three times a week for a total duration of three weeks. Rat IgG2a, rat IgG1, Armenian hamster IgG, and mouse IgG1 isotype control antibodies were purchased from Bio X Cell (West Lebanon, NH, USA). Isotype control antibodies were administered to control mice in correspondence with the original antibodies. From one-week post-infection, 20 µg/mouse of AM80 (Sigma-Aldrich, St. Louis, MO, USA) was orally administered. AM80 was diluted in PBS and administered three times a week for a total duration of three weeks.

### Antibodies and flow cytometry

Flow cytometric analysis of immune cells was conducted using the subsequent antibodies. LIVE/DEAD Fixable Near-IR Dead Cell Stain Kit, Green Dead Cell Stain Kit, and Aqua Dead Cell Stain Kit were purchased from Molecular Probes (Carlsbad, CA, USA). Brilliant violet (BV) 605-conjugated mAb against Thy1.2 and CD19; Allophycocyanin (APC)-conjugated mAb against CD45R (B220); BV 421-conjugated mAb against CD45; APC-R700-conjugated mAb against Siglec-F; Fluorescein isothiocyanate (FITC)-conjugated mAb against CD95 (FAS); BV 786-conjugated mAb against CXCR2; and V450- conjugated mAb against Ly6G were purchased from BD Bioscience (San Jose, CA, USA). Phycoerythrin (PE)-conjugated mAb against CXCR2, CD64, IFN-γ, and GL7; Peridinin chlorophyll (PerCP)-Cy5.5-conjugated mAb against CD11b; APC-Cy7-conjugated mAb against MHC-II; Alexa fluor 700-conjugated mAb against CD62L; and PE-Dazzle conjugated mAb against CD11c were purchased from Biolegend (San Diego, CA, USA). Surface and intracellular staining processes were executed according to the referenced study [87].

### Preparation of single cell suspensions and immune cell analysis

Four weeks post-infection, the mice were euthanized via inhalation of carbon dioxide and then underwent an autopsy. The mouse lungs were perfused before undergoing flow cytometry analysis. Using the methods described in previous studies [61,88], single-cell suspensions were obtained from the entire lung tissue. These cells were then stained with the mAbs mentioned in the previous section and subjected to flow cytometry analysis.

### Measurement of cytokines

For the analysis of lung cytokines, lung lysates from the autopsied mice were collected, homogenized, dissolved into PBS, and then stored in a -80˚C freezer. For the analysis of T cell cytokines, lung single cell suspensions were stimulated with two different Mtb antigens, PPD and ESAT-6 according to the referenced studies [87,88]. To detect cytokines, sandwich enzyme-

linked immunosorbent assay (ELISA) and LEGENDplex kits were utilized. Briefly, mouse G-CSF and IL-1α ELISA kits were purchased from R&D Systems (Minneapolis, MN, USA). Mouse IFN-γ, IL-1β, IL-17A, and IL-5 ELISA kits were purchased from Invitrogen (San Diego, CA, USA). Mouse IL-10 ELISA kit, LEGENDplex mouse B cell panel (for the detection of TNF-α, IL-4, IL-6, BAFF, and TGF-β), and LEGENDplex mouse inflammation panel (for the detection of IL-1α, IL-1β) were purchased from Biolegend (San Diego, CA, USA).

## Quantification of lung inflammation and mycobacterial CFU

The lungs and spleens of Mtb infected mice were harvested four weeks post-infection. The right superior lobes of the lungs were kept in 10% formalin overnight for preservation and later embedded in paraffin. To perform histopathologic analysis, the lungs were sectioned at 4–5 μm and stained with hematoxylin and eosin (H&E). To quantify pulmonary inflammation in each mouse, we employed Adobe Photoshop (Adobe, San Jose, California) and ImageJ (National Institutes of Health, USA) programs, following a previously described reference [89]. The complete lung lesion image was isolated using Adobe Photoshop and saved as a separate file. Another image, identical in size to the original lung image but filled with black, was also saved as a separate file. These images were then opened in ImageJ. By using the 'Split Channels' function, the image from the green channel was processed to calculate the size of the area showing green positivity, which was originally purple-blue due to H&E staining. The black-colored image was analyzed through the green channel in the same manner to calculate the size of the entire lung lesion. The 'inflamed area (%)' value, which indicates the proportion of inflamed tissue relative to the entire lung lesion, was computed by dividing the green positivity values of the first image by the green positivity values of the black-colored image. Subsequently, the actual size of the inflamed lesion was calculated by determining the pixel values in Adobe Photoshop. The pixel value corresponding to a single millimeter was calculated based on a 2mm scale bar included in each images. The pixel value for the entire lung lesion was also enumerated, and the value was divided by the square of the pixel value representing one millimeter to quantify the size (mm$^2$) of each lung lesion. By multiplying the inflamed area (%) values by the total lung sizes (mm$^2$) for each sample, we were able to precisely enumerate the exact size of each inflamed area (mm$^2$) (**S11 Fig**). To enumerate mycobacterial growth, lung and spleen tissues were plated onto Middlebrook 7H10 agar (Becton Dickinson, Franklin Lakes, NJ, USA) after homogenization, following preparation methods from previous studies [61,90]. After a four-week incubation period at 37˚C, mycobacterial colonies were counted.

## Neutrophil purification and assessment of mycobacterial permissiveness

MACS magnetic cell sorting kit including anti-Ly6G magnetic beads and LS MACS columns (Miltenyi Biotec, Bergisch Gladbach, Germany) was used to enrich Ly6G$^+$ cells from the lungs of Mtb-infected mice. The manufacturer's protocols were followed. After separating Ly6G$^+$ lung cells, the cells were treated with Triton X-100 (Sigma-Aldrich, St. Louis, MO, USA) and plated onto Middlebrook 7H10 agar (Becton Dickinson, Franklin Lakes, NJ, USA). After a three-week incubation period at 37˚C, mycobacterial colonies were counted.

## Statistical analyses

The results were presented as the mean ± standard deviation (SD). To analyze the significance of differences between two selected groups, One-way ANOVA and Mann-Whitney U test were conducted using GraphPad Prism version 8 for Windows (GraphPad Software, La Jolla, CA, USA, www.graphpad.com). The statistical significance was determined using the following definitions: *n.s.*: not significant, *$p < 0.05$, and **$p < 0.01$. Correlation analysis and

unpaired t-test were conducted using GraphPad Prism, and the correlation coefficients and p-values were reported for each graph. Flowjo V10 (Flowjo, Ashland, OR, USA) was employed to perform t-distributed stochastic neighbor embedding (t-SNE) analysis.

## Supporting information

**S1 Fig. The gating strategy for immune cell populations and the kinetics time-course data of pulmonary CFUs and neutrophils are provided. (A)** Pulmonary CFUs of female WT and *Nox2*$^{-/-}$ mice were enumerated in a time-course dependent manner. six-week old female WT and *Nox2*$^{-/-}$ mice (n = 5 per group) were aerosol infected with Mtb K strain. Mycobacterial CFUs in the lungs of Mtb infected mice were calculated at 0, 2, 3, 4, 8, and 12 weeks post-infection. Initial CFU = 373. The experiment was conducted once. **(B)** Pulmonary neutrophil counts were calculated at 0, 2, 3, 4, 8, and 12 weeks post-infection. The experiment was conducted once. The significance of differences was determined, using the One-way ANOVA and Mann-Whitney U test. *n.s.*, not significant. $^{*}p < 0.05$. $^{**}p < 0.01$. (C) The gating strategy for immune cell populations is presented as flow cytometry plots. Firstly, Live cells stained with the LIVE/DEAD Aqua Dead Cell Stain Kit were gated among single cells. CD45$^{+}$ immune cells were then gated in order to investigate lung immune cell compositions. Among CD45$^{+}$ immune cells, CD19$^{+}$CD90.2$^{+}$ SSC-A$^{lo}$ lymphocytes and CD19$^{-}$CD90.2$^{-}$SSC-A$^{hi}$ myeloid cells were discriminated. Lymphocytes consisted of CD90.2$^{+}$ T cells, CD19$^{+}$B220$^{+}$ B cells, and CD19$^{+}$B220$^{+}$GL7$^{+}$FAS$^{+}$ germinal center B cells. Myeloid cells consisted of CD11b$^{+}$Ly6G$^{+}$ neutrophils, CD11c$^{+}$Siglec-F$^{+}$ alveolar macrophages, CD11b$^{+}$CD64$^{+}$ macrophages, and CD11c$^{+}$MHC-II$^{+}$ dendritic cells.
(TIF)

**S2 Fig. Pulmonary T cells of male *Nox2*$^{-/-}$ mice maintained strong responses against mycobacterial antigens.** Live lung cell suspensions were cultured with or without the mycobacterial antigens ESAT-6 and PPD for 8 hours. **(A)** Lung IFN-γ levels and **(B)** Lung IL-17A levels after mycobacterial antigen stimulus are presented in bar graphs. **(C)** IFN-γ positive CD90.2$^{+}$ CD4$^{+}$ CD44$^{+}$ CD62L$^{+}$ effecter T cell populations are presented in bar graphs and flow cytometry plots. The data are presented as the mean ± SD of six mice in each group. The significance of differences was determined, using the One-way ANOVA test. *n.s.*, not significant. $^{**}p < 0.01$.
(TIF)

**S3 Fig. Alveolar macrophages, macrophages, and dendritic cells were reduced in the lungs of Mtb infected male *Nox2*$^{-/-}$ mice.** Pulmonary **(A)** CD11c$^{+}$Siglec-F$^{+}$ alveolar macrophages, **(B)** CD11b$^{+}$CD64$^{+}$ macrophages, and **(C)** CD11c$^{+}$MHCII$^{+}$ dendritic cell populations of Mtb-infected mice at four weeks post-infection. The percentages of each immune cell among lung CD45$^{+}$ cells and total cell counts are presented in bar graphs. The data are presented as the mean ± SD of six mice in each group. The significance of differences was determined, using the One-way ANOVA test. $^{**}p < 0.01$.
(TIF)

**S4 Fig. TB severity featured strong correlation to increase of lung neutrophils and decrease of lung B cells.** The percentages of **(A)** neutrophils, **(B)** T cells, and **(C)** B cells among lung CD45$^{+}$ cells were individually correlated with inflamed lung percentages, inflamed lung area, and lung bacterial loads of each mouse and presented in correlation graphs. Additionally, **(D)** the ratio of pulmonary B cells to neutrophils (B:N ratio) was also individually correlated with inflamed lung percentages, inflamed lung area, and lung bacterial loads of each mouse and presented in correlation graphs. The significance of differences was determined by unpaired t-

test and correlation analysis, featuring r and p values of each correlation.
(TIF)

**S5 Fig. BM and lung neutrophil profiles of uninfected WT and *Nox2*-/- mice are illustrated.** Neutrophil counts and percentages from the lungs and bone marrows of uninfected WT and *Nox2*-/- mice are provided (n = 4 per group). **(A)** The percentages of neutrophils among lung CD45+ cells and total cell counts are presented in bar graphs, along with flow cytometry plots. **(B)** Total cell counts and the percentages of CXCR2$^{lo}$CD62L$^{lo}$ immature neutrophils and CXCR2$^{hi}$CD62L$^{hi}$ mature neutrophils among total lung neutrophils are presented in bar graphs, along with flow cytometry plots. **(C)** The percentages of neutrophils among CD45+ bone marrow cells and total cell counts are presented in bar graphs, along with flow cytometry plots. **(D)** Total cell counts and the percentages of CXCR2$^{lo}$CD62L$^{lo}$ immature neutrophils and CXCR2$^{hi}$CD62L$^{hi}$ mature neutrophils among total bone marrow neutrophils are presented in bar graphs, along with flow cytometry plots. The data are presented as the mean ± SD of four mice in each group. The significance of differences was determined using the One-way ANOVA test and Mann-Whitney-U test. *n.s.*, not significant. *$p < 0.05$. **$p < 0.01$. significant.
(TIF)

**S6 Fig. Immature neutrophils were the most permissive cells to mycobacterial infection among phagocytes infected with YFP+ Mtb. (A)** Experimental design for the *in vivo* enumeration of permissiveness in phagocytes. Male WT and *Nox2*-/- mice (n = 5 per group) were aerosol infected with YFP-expressing Mtb K strain. At four weeks post-infection, all mice were autopsied, and permissiveness of neutrophils were analysed via flow cytometry (indicated by red arrow). Initial CFU = 742. **(B)** The percentage of neutrophils among YFP+ cells are presented in bar graphs, along with flow cytometry plots. The number of YFP+ neutrophils and MFI values of YFP in concatenated neutrophils (10$^3$ cells) are presented in bar graphs. **(C)** Total cell counts, percentages among YFP+ neutrophils, and MFI values of YFP (per 10$^3$ concatenated cells) in CXCR2$^{lo}$CD62L$^{lo}$ immature neutrophils and CXCR2$^{hi}$CD62L$^{hi}$ mature neutrophils are presented in bar graphs, along with flow cytometry plots. Total cell counts, percentages among YFP+ cells, and MFI values of YFP (per 10$^3$ concatenated cells) in **(D)** CD11c+Siglec-F+ alveolar macrophages and **(E)** CD11b+CD64+ recruited macrophages are presented in bar graphs, along with flow cytometry plots. The experiment was conducted once. The data are presented as the mean ± SD of five mice in each group. The significance of differences was determined using the One-way ANOVA test and Mann-Whitney-U test. *n.s.*, not significant. *$p < 0.05$. **$p < 0.01$.
(TIF)

**S7 Fig. AM80 treatment did not significantly alter T cells responses against mycobacterial antigens.** Live lung cell suspensions were cultured with or without mycobacterial antigens ESAT-6 and PPD for 8 hours. **(A)** Lung IFN-γ levels and **(B)** Lung IL-17A levels after mycobacterial antigen stimulus are presented in bar graphs. **(C)** IFN-γ positive CD90.2+ CD4+ CD44+ CD62L+ effecter T cell populations are presented in bar graphs and flow cytometry plots. The data are presented as the mean ± SD of five mice in each group. The significance of differences was determined, using the One-way ANOVA test and Mann-Whitney test. *n.s.*, not significant. *$p < 0.05$. **$p < 0.01$.
(TIF)

**S8 Fig. Neutralization of IFN-γ and IL-17A exacerbated TB pathogenesis of female *Nox2*-/- mice. (A)** Experimental design for *in vivo* neutralization of IFN-γ and IL-17A in Mtb infected mice. six-week old female *Nox2*-/- mice (n = 5 per group) were aerosol infected with Mtb K

strain. Starting from two weeks post-infection, 200 µg of anti-IFN-γ mAb or 250 µg of anti-IL-17A mAb was intraperitoneally administered to each mouse three times a week (indicated by blue bars). At four weeks post-infection, all mice were autopsied, and immunological analysis, bacterial counting, and histopathological analysis were conducted (indicated by red arrow). Initial CFU = 225. **(B)** Mycobacterial CFUs in the lungs and spleens of each group at four weeks post-infection were analyzed by calculating the number of colonies and presented in bar graphs. **(C)** H&E staining was performed on the superior lobes of the right lung at four weeks post-infection to visualize the gross lung pathology. The inflamed area of the H&E-stained samples was quantified in terms of percentage and square millimeters and presented in bar graphs. **(D)** IFN-γ, IL-1α, IL-1β, IL-17A, IL-6, TNF-α, G-CSF, and BAFF levels in Mtb-infected mouse lung lysates were measured by ELISA and LEGENDplex **(E)** IL-10, IL-4, IL-5, and TGF-β levels in Mtb-infected mouse lung lysates were measured by ELISA and LEGEN-Dplex The cytokine levels are presented in bar graphs. Pulmonary **(F)** CD11b$^+$Ly6G$^+$ neutrophil, **(G)** CD11b$^+$Ly6G$^+$CXCR2$^{lo}$CD62L$^{lo}$ immature neutrophil, and CD11b$^+$Ly6G$^+$CXCR2$^{hi}$CD62L$^{hi}$ mature neutrophil populations of Mtb-infected mice at four weeks post-infection. The percentages of each immune cell among lung CD45$^+$ cells and total cell counts are presented in bar graphs, along with flow cytometry plots. The experiment was conducted once. The data are presented as the mean ± SD of five mice in each group. The significance of differences was determined, using the One-way ANOVA test. *n.s.*, not significant. $^*p < 0.05$. $^{**}p < 0.01$.
(TIF)

**S9 Fig. Neutralization of IL-6, IL-1α, and IL-1β exacerbated TB pathogenesis of male *Nox2*<sup>-/-</sup> mice. (A)** Experimental design for *in vivo* neutralization of IL-6, IL-1α, and IL-1β in Mtb infected mice. six-week old male *Nox2*<sup>-/-</sup> mice (n = 5 per group) were aerosol infected with Mtb K strain. Starting from one week post-infection, 400 µg of anti-IL-6 mAb or 200 µg of anti-IL-1α mAb or 200 µg of anti-IL-1β mAb was intraperitoneally administered to each mouse three times a week (indicated by blue bars). At four weeks post-infection, all mice were autopsied, and immunological analysis, bacterial counting, and histopathological analysis were conducted (indicated by red arrow). Initial CFU = 330. **(B)** Mycobacterial CFUs in the lungs and spleens of each group at four weeks post-infection were analyzed by calculating the number of colonies and presented in bar graphs. **(C)** H&E staining was performed on the superior lobes of the right lung at four weeks post-infection to visualize the gross lung pathology. The inflamed area of the H&E-stained samples was quantified in terms of percentage and square millimeters and presented in bar graphs. **(D)** IFN-γ, IL-1α, IL-1β, IL-17A, IL-6, TNF-α, G-CSF, and BAFF levels in Mtb-infected mouse lung lysates were measured by ELISA and LEGENDplex **(E)** IL-10, IL-4, IL-5, and TGF-β levels in Mtb-infected mouse lung lysates were measured by ELISA and LEGENDplex The cytokine levels are presented in bar graphs. Pulmonary **(F)** CD11b$^+$Ly6G$^+$ neutrophil, **(G)** CD11b$^+$Ly6G$^+$CXCR2$^{lo}$CD62L$^{lo}$ immature neutrophil, and CD11b$^+$Ly6G$^+$CXCR2$^{hi}$CD62L$^{hi}$ mature neutrophil populations of Mtb-infected mice at four weeks post-infection. The percentages of each immune cell among lung CD45$^+$ cells and total cell counts are presented in bar graphs, along with flow cytometry plots. The experiment was conducted once. The data are presented as the mean ± SD of five mice in each group. The significance of differences was determined, using the One-way ANOVA test. *n.s.*, not significant. $^*p < 0.05$. $^{**}p < 0.01$.
(TIF)

**S10 Fig. Blockade of IL-1R exacerbated TB pathogenesis of *Nox2*<sup>-/-</sup> mice. (A)** Experimental design for *in vivo* blockade of IL-1R in Mtb infected female *Nox2*<sup>-/-</sup> mice. six-week old female *Nox2*<sup>-/-</sup> mice (n = 5 per group) were aerosol infected with Mtb K strain. Starting from one

week post-infection, 200 μg of anti-IL-1R mAb was intraperitoneally administered to each mouse three times a week (indicated by blue bars). At four weeks post-infection, all mice were autopsied, and bacterial counting and histopathological analysis were conducted (indicated by red arrow). Initial CFU = 300. **(B)** Mycobacterial CFUs in the lungs and spleens of each group at four weeks post-infection were analyzed by calculating the number of colonies and presented in bar graphs. **(C)** H&E staining was performed on the superior lobes of the right lung at four weeks post-infection to visualize the gross lung pathology. The inflamed area of the H&E-stained samples was quantified in terms of percentage and square millimeters and presented in bar graphs. **(D)** Pulmonary CD11b$^+$Ly6G$^+$ neutrophil populations of Mtb-infected mice at four weeks post-infection. The percentages of neutrophils among lung CD45$^+$ cells and total cell counts are presented in bar graphs, along with flow cytometry plots. **(E)** Experimental design for *in vivo* blockade of IL-1R in Mtb infected male *Nox2*$^{-/-}$ mice. six-week old male *Nox2*$^{-/-}$ mice (n = 5 per group) were aerosol infected with Mtb K strain. Starting from two weeks post-infection, 200 μg of anti-IL-1R mAb was intraperitoneally administered to each mouse three times a week (indicated by blue bars). At four weeks post-infection, all mice were autopsied, and bacterial counting and histopathological analysis were conducted (indicated by red arrow). Initial CFU = 342. **(F)** Mycobacterial CFUs in the lungs and spleens of each group at four weeks post-infection were analyzed by calculating the number of colonies and presented in bar graphs. **(G)** H&E staining was performed on the superior lobes of the right lung at four weeks post-infection to visualize the gross lung pathology. The inflamed area of the H&E-stained samples was quantified in terms of percentage and square millimeters and presented in bar graphs. **(H)** Pulmonary CD11b$^+$Ly6G$^+$ neutrophil populations of Mtb-infected mice at four weeks post-infection. The percentages of neutrophils among lung CD45$^+$ cells and total cell counts are presented in bar graphs, along with flow cytometry plots. The experiments were conducted once. The data are presented as the mean ± SD of five mice in each group. The significance of differences was determined, using the Mann-Whitney test. *n. s.*, not significant. *$p < 0.05$. **$p < 0.01$.
(TIF)

**S11 Fig. Quantification method of lung inflammation is briefly described. (A)** As outlined in the Materials and Methods section, the inflamed area of the H&E-stained lung samples were quantified in terms of percentage and square millimetres. The values were calculated using Adobe Photoshop and ImageJ programs. New image files were created to determine the percentage of the inflamed (purple-stained) lesions, and the actual size of both total lung lesions and inflamed lesions was quantified in mm$^2$.
(TIF)

## Acknowledgments

We appreciate the Medical Illustration & Design(MID) team, a member of Medical Research Support Services of Yonsei University College of Medicine, for their excellent support with medical illustration.

## Author Contributions

**Conceptualization:** Eunsol Choi, Hong-Hee Choi, Sung Jae Shin.

**Data curation:** Eunsol Choi, Kee Woong Kwon, Hagyu Kim.

**Formal analysis:** Eunsol Choi.

**Funding acquisition:** Kee Woong Kwon, Sung Jae Shin.

**Investigation:** Eunsol Choi, Hong-Hee Choi, Kee Woong Kwon, Hagyu Kim.

**Methodology:** Eunsol Choi, Hong-Hee Choi, Kee Woong Kwon, Hagyu Kim, Ji-Hwan Ryu, Jung Joo Hong.

**Project administration:** Jung Joo Hong, Sung Jae Shin.

**Resources:** Ji-Hwan Ryu, Jung Joo Hong, Sung Jae Shin.

**Software:** Eunsol Choi.

**Supervision:** Sung Jae Shin.

**Validation:** Eunsol Choi.

**Visualization:** Eunsol Choi.

**Writing – original draft:** Eunsol Choi.

**Writing – review & editing:** Eunsol Choi.

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
