## [Decision Letter · Decision Letter 0]

26 Jan 2024

Dear Dr. Shin,

We apologize for the delay in evaluating your manuscript, "Permissive lung neutrophils facilitate tuberculosis immunopathogenesis in male phagocyte NADPH oxidase-deficient mice".  Given the initial comments we received, we felt that it was particularly important to secure three thorough reviews, and this delayed our decision.   In light of these reviews (below this email), we would like to invite the resubmission of a significantly-revised version that takes into account the reviewers' comments.

You will note that the reviewers appreciated the importance of the topic and noted a number of interesting observations.  They also shared several concerns that would require additional experimental work to alleviate.  We cannot make any decision about publication until we have seen the revised manuscript and your response to the reviewers' comments. Your revised manuscript is also likely to be sent to reviewers for further evaluation.

Sincerely,

Christopher M. Sassetti

Academic Editor

PLOS Pathogens

Michael Otto

Section Editor

PLOS Pathogens

Kasturi Haldar

Editor-in-Chief

PLOS Pathogens

orcid.org/0000-0001-5065-158X

Michael Malim

Editor-in-Chief

PLOS Pathogens

orcid.org/0000-0002-7699-2064

Reviewer's Responses to Questions

**Part I - Summary**

Reviewer #1: In this manuscript “Permissive lung neutrophils facilitate tuberculosis immunopathogenesis in male phagocyte NADPH oxidase-deficient mice,” Choi et al. explore the effects of NOX2 deficiency on Mycobacterium tuberculosis infection. They find that NOX2 deficiency leads to increased lung inflammation and bacterial burdens, which is associated with increased inflammation and immature neutrophilia, all with a greater difference in male mice. They next go on to evaluate the potential mechanistic underpinning for these observations, showing that this susceptibility in the absence of NOX2 is in part dependent on neutrophils, and associated with immature neutrophils and G-CSF, further this can be ameliorated with a drug which stimulates the retinoic acid receptor. Overall, this study highlights a key role for NOX2 in the regulation of pathology during Mycobacterium tuberculosis infection and uses a diverse array of immunologic tools to characterize its effects. Enthusiasm for this manuscript is significantly tempered by three key points: 1) Even in the presence of anti-Ly6G treatment, NOX2 deficiency results in a nearly 1 log increase in CFU (fig 4B, green vs blue bars), suggesting a significant contribution of non-neutrophil factors. The authors do not address this, instead their interpretation is that neutrophils account for all of differences seen in NOX2 deficiency; 2) Related to this, NOX2 deficiency has a significant effect on other phagocytes, including macrophages, which are examined only superficially; 3) At the relatively late timepoint evaluated, there are already significant differences in bacterial burden and it is difficult to discern cause versus effect. This caveat could be mitigated by examining earlier timepoints. Lastly, it is not clear whether any of the experiments were repeated, which is critical to evaluate these findings, especially in light of the difference of these findings with published tuberculosis literature on gp91-/- mice (ref 29 in this publication), which as the authors state could be due to strain differences.

Reviewer #2: In this manuscript by Choi et al the investigators examine differences in role of phagocyte oxidase mediated protection in male and female mice. They find that male Nox2-/- mice have higher bacterial burden following infection with a virulent Strain K, while female mice do not. They find increased immature neutrophil recruitment and lung damage at 4 weeks post-infection that is associated with higher Mtb levels and a hyperinflammatory response. Depleting neutrophils in male mice reversed the increased CFU and hyperinflammation while IL1 blockade did not. In contrast, treatment with AM80 and anti-G-CSF lowered bacterial levels in male Nox2-/- mice and reversed the inflammatory state. The authors conclude that increased G-CSF in male mice specifically results in the recruitment of immature neutrophils that drive inflammation and increased bacterial burden. The paper is generally clear and the experiments are mostly well controlled with a few caveats. Several of the key observations including the role of increased neutrophil recruitment and the role of immature neutrophils in driving disease have previously been described. Furthermore, several of the data presented appear to conflict with other previous reports using various knockouts of distinct subunits of the NADPH phagocyte oxidase, yet the reasons for these discrepancies are not fully understood/examined. Without further experiments or controls this leaves some of the conclusions not fully supported by the data as is.

Reviewer #3: Choi and colleagues in this study have characterized the lung neutrophils phenotype in female and male Mtb infected wild type (WT) and Nox2-KO mice; and showed that increased immature CXCR2lo CD62Llo Ly6G+ lung neutrophils were associated with enhanced TB pathology in male Nox2-kO mice. They performed depletion experiments using antibodies against various cytokines and found that GCSF-driven differentiation of immature neutrophils was responsible for exacerbated TB pathology in male Nox2-KO mice. These kinds of studies are important to dissect the pathologic and protective role of neutrophils in TB and possible in other infectious diseases. What is the phenotype of depleting immature neutrophils in WT mice? Does depleting GCSF in WT mice has an effect on TB pathology? These biological questions and significant technical issues lowered the interest of this reviewer.

**Part II – Major Issues: Key Experiments Required for Acceptance**

Reviewer #1: Given the large differences in bacterial burdens, it is difficult to know which changes in immune cell number/proportion and increased inflammatory cytokines seen in the male NOX2 deficient mice are reflective of increased bacterial burdens, or responsible for driving the pathology. To gain better insight into the mechanistic determinants for NOX2-driven pathology, the authors could evaluate responses at an earlier timepoint, when there is no (or less) discrepancy in bacterial burdens. Also, why is this data different from the published gp91-/-, as above do the authors think this is solely related to bacterial strain, or potentially could infectious dose also influence these responses?

The methods outlined for determining extent of inflammation are not sufficient to understand what this is measuring. The listed reference also does not have enough detail (overall there is not enough detail in the methods). In the text, authors refer to “necrotic inflammation” whereas the figure axes are labeled “inflamed area,” which are quite different. Also, from thin tissue sections of a region of interest, I’m not sure how useful the absolute area of inflammation is, the percent inflammation controls for the size of the section, which is the most relevant to analyze. Overall, the images aren’t at a sufficient resolution to evaluate in the uploaded tif files. Perhaps the authors could include images from all evaluated animals in a supplemental figure?

The authors propose that immature lung neutrophils are permissive, but never show this specifically on immature neutrophils in the setting of NOX2 deficiency. In an above (or related) experiment this could be shown by sorting and plating immature and mature neutrophils. Otherwise, the authors should make a statement more generally about neutrophils, which they have shown have higher bacterial burdens.

AM80 treatment has a relatively modest effect on the absolute number of neutrophils present. Given the mechanism proposed, is there a difference in the percentage of immature neutrophils? Also, the proportion of neutrophils does not seem to change, which is similar to other cell populations. Are the absolute numbers of all cell populations decreased? If so, it doesn’t seem fair to conclude that immature neutrophils are the key cell population, if many other cell populations changed as well. Given the effects of ATRA treatment in reference 58, were responses of T cells and macrophages evaluated further?

Reviewer #2: Major Comments:

1. Several previous papers cited by the authors show that infection of Phox deficient mice does not result in major changes to CFU over time in contrast to what is shown. The authors suggest this is due to difference strains however, but this is not directly tested. The authors only examine a single time point in their studies (4 weeks post infection) and do not examine the kinetics of bacterial growth over time which were shown to have some variability during infection or the impact on survival long term. The authors use a wild type C57BL6N background which also differs from previous studies. It is unclear from the methods whether the Nox2-/- mouse is also on the C57BL6N or the C57BL6J background. This should be made clearer as a potential confounder.

2. For several of the treatment studies including AM80, G-CSF, IL1a, IL1b, IL6 there are no wild type controls to understand changes in control mice. This is particularly important for AM80 which the authors conclude is driving better neutrophil maturation and control, however, this could also be directly acting on infected macrophages to lower bacterial burden as has been shown previously in human macrophages. Without knowing if AM80 is specifically driving neutrophil mediated control these conclusions are not supported by the current data. The authors also conclude the changes in Male Nox2-/- mice are not due to IL1a or IL1b but a co-treatment is not done even though these cytokines can act redundantly. Furthermore, it is known that IL1 signaling is important for early control of Mtb infection yet depletion beings very early following aerosol infection which again, may be driving changes in macrophage bacterial levels that then amplify differences in neutrophil responses. This is not addressed through experiments.

3. The majority of the statistics seem to use the Mann-Whitney U test which from my understanding is a comparison between two groups and in several experiments there are more than two comparisons being made. A more appropriate test should be used for these experiments to account for multiple comparisons. Furthermore, in the figure legends it is unclear how many replicate experiments were done for each data panel making it difficult to judge the reproducibility of the results.

4. The methods in general are not sufficient for reproducible studies. While previous studies can be cited the manuscript should contain enough information for readers to fully understand the results. For example, how was the Strain K cultured? Is it PDIM positive? For the aerosol infections, was the day 0 lung CFU value calculated for each experiment (if so this should be explicitly written in the figure legends)? Were there differences in these values between male and female mice that may drive downstream phenotypes? These data are important for the readers to understand all the parameters of the experiments.

Reviewer #3: 1. Thomas et al. 2023 have earlier demonstrated that gp91phox-deficient mice displayed enhanced inflammation and lung neutrophil infiltration after Mtb challenge, which were mediated by IL-1. In this context a Fig 1 and Fig 2 of this manuscript is just repeating what was published before. They can be merged into oone figure. Having said that the current manuscript though brings the angle of sex, which is interesting.

2. Immature neutrophils and pathology: Regarding the observation that Nox2-KO mice exhibited higher levels of immature lung neutrophils, inflammation and mycobacterial burden compared to female Nox2-KO mice and WT mice after Mtb challenge. Is this due to the inherent nature of higher immature neutrophils in the lung of mice at the time of infection, or those increased immature neutrophils arrive only upon infection (granulopoisis in bone marrow). Thus, It is important to know the neutrophil phenotype (possibly in lung and bone marrow) in uninfected female / male WT and Nox2-KO mice.

3. In Fig S4 authors showed that “The loss of pulmonary B cells showed a significant negative correlation with neutrophil infiltration”. The negative correlation of neutrophils and B cells was again depicted in neutrophil depletion experiments in Fig. 5. Authors have not provided much explanation on this; it will be good to discuss this aspect in detail and its impact on TB pathogenesis.

4. For immature neutrophils I would suggest including Evrard et al., 2018 Immunity article as reference; This was one of the first seminal paper to characterize different neutrophil subtypes.

5. Authors mentioned that “CXCR2loCD62Llo immature lung neutrophils were phenotypically heterogeneous from CXCR2hiCD62Lhi mature lung neutrophils”. This is not clear. Which figure panel shows this? Was any sub-gating of these immature and mature neutrophils (using other markers) was performed? How the heterogenity was measured / assessed?

6. Fig 5E/6H – Not sure what kind of information tsne plots are adding in this panel. Same goes for other figures where tsne plots have been shown. For example in Fig 6H - I can see many green colour clusters (all are T cells), but what is what; which is CD4, CD8, MAIT, etc etc. If the authors do not have that granular information, avoid including tsne plots. This will be very confusing for the readers.

7. Fig 6 – Does AM80 treatment also ameliorate TB pathology / bacterial load in wild type mice? Why this experiment was performed? Good to make it more clear.

8. Fig S5 – I could not get whether this experiment (depletion of IFNg and IL17A) was performed with male or female mice. In the text its mentioned male Nox2-KO mice however in the Fig S5 legend its mentioned female Nox2-KO mice.

9. Does depleting GCSF in wild type mice also has an effect on lung immature neutrophils, bacillary load and pathology? This is very important to investigate as this will generalize the pathological role of immature neutrophils in TB; In my view this study will then have a more clinical relevance.

10. Figure legend – please include how many times each experiment was repeated.

11. Statistical analysis – I am surprised to see that authors have used Mann-Whitney test, though they have more than 2 groups in maximum experimental settings. This is not correct. Whenever there are more than 2 groups ANOVA should be used.

12. Depletion experiments - I could not find the info (neither in method section nor in figure legend) on the control Ab used in all depletion studies. Were the control mice receive any antibody?

**Part III – Minor Issues: Editorial and Data Presentation Modifications**

Reviewer #1: The sex differences in NOX2 deficiency are very interesting. However, it may appear that there could be a difference in female mice as well, potentially with lung CFU and neutrophil number. It is not clear from the plots or text if a statistical test has been applied to these differences, which should be addressed on the figures and in the text as appropriate. Related to this: the plots for CFU and inflamed area should include points for individual mice, as there are in every other figure panel.

Is the same clone of antibody used to deplete neutrophils as it is to detect neutrophils in figures 4 and 5? If so, it is important to see a method of neutrophil quantification that does not rely on Ly6G. This may be able to be accomplished with existing flow markers. As a minor note in this figure, it is a circular argument to say that neutrophil depletion restores the balance between neutrophils and lymphocytes. What does it do to the absolute numbers of lymphocytes?

Would the authors address why there is so much IFNg, IL1, IL6, and IL17 detected in the setting of the respective cytokine depletion experiments? Was there another method implemented to show that neutralization was effective?

G-CSF is a nice way addition to specifically target neutrophils. It can have multiple effects in addition to granulopoiesis, including on mature neutrophil function and migration into tissues. Evaluating these mechanisms may be outside of the scope of this study, however the authors should address that it might not be as simple as they propose. Also to the above point, did the authors perfuse lungs to ensure that most cells evaluated were within the lung parenchyma? It does not appear that intravascular antibody labeling was completed.

Displaying cell proportions as pie charts makes is impossible to evaluate statistical relationships. Please include statistical measurements for key comparisons,

When the labels of the graphs are at the bottom require a lot of scrolling to evaluate all panels. Please include labels under each graph.

The key for supplemental figure S6 is missing the anti-IL6 only group.

Reviewer #2: (No Response)

Reviewer #3: 1. Antibodies – Please provide the clone information.

2. This article needs an English editing.

3. Enumerating CFU in isolated neutrophils – Did the cells were counted after MACS sorting?

4. References – Some of the references are not needed can be removed, for example 47 and 49. There are other examples in the text.

5. The way four groups are depicted in Fig 3B is very confusing. Please relabel them.

PLOS authors have the option to publish the peer review history of their article (what does this mean?). If published, this will include your full peer review and any attached files.

Reviewer #1: No

Reviewer #2: No

Reviewer #3: No
---

## [Decision Letter · Decision Letter 1]

13 May 2024

Dear Dr. Shin,

Thank you very much for submitting your manuscript "Permissive lung neutrophils facilitate tuberculosis immunopathogenesis in male phagocyte NADPH oxidase-deficient mice" for consideration at PLOS Pathogens. As with all papers reviewed by the journal, your manuscript was reviewed by members of the editorial board and by several independent reviewers. In light of the reviews (below this email), we would like to invite the resubmission of a significantly-revised version that takes into account the reviewers' comments.

The reviewers appreciated your revisions and additional experimental data, but note a number of points that remain to be addressed.  While we always endeavor to avoid multiple rounds of review/revision, these remaining issues are significant and must be addressed before publication.  Specifically, reviewers 1 and 2 note several issues to address by revising the text and/or providing additional flow cytometry data.  We recognize that these revisions may be significant, but trust that this constructive will produce a more impactful manuscript.  

We cannot make any decision about publication until we have seen the revised manuscript and your response to the reviewers' comments. Your revised manuscript is also likely to be sent to reviewers for further evaluation.

Sincerely,

Michael Otto

Section Editor

PLOS Pathogens

Michael Otto

Section Editor

PLOS Pathogens

Michael Malim

Editor-in-Chief

PLOS Pathogens

orcid.org/0000-0002-7699-2064

Reviewer's Responses to Questions

**Part I - Summary**

Reviewer #1: Choi et al. have done a nice job generating and providing additional data, and making progress in addressing the reviewers’ comments in the following areas. First, the analysis of the permissiveness of immature neutrophils using a fluorescent Mtb strain is valuable and supports the authors’ conclusions. Second, showing the effects of G-CSF and AM80 in WT mice, and other necessary comparisons such as analysis of uninfected mice is a very welcome addition. Lastly, the figure clarity and readability has improved.

However, this reviewer still has substantial concerns about many of the conclusions reached by the authors in this manuscript. The article does highlight a major role for neutrophil-driven pathology during Mtb infection in the setting of NOX2 deficiency. However, it is unlikely that this is the only process affected by NOX2 deficiency, and non-neutrophil alternative explanations are only superficially addressed / dismissed. Additionally, the AM80 treatment results in WT mice, together with other data presented in the paper, call into question the interpretation that this treatment acts primarily through neutrophils. Lastly, the fact that findings were observed at high doses with a highly virulent strain, when others have found different results with different strains, could either suggest interesting biology and/or limit the broad applicability of these findings. Given this, the article as written needs some reframing and significant revision of the conclusions drawn from these studies.

Reviewer #2: In this revised manuscript by Choi et al, the authors examine an increase in CFU in male Nox2 mice at 4 weeks post infection and if blocking G-CSF can prevent the neutropenia seen in these animals. The revised manuscript aims to address reviewer concerns, and some of the data are informative, including the YFP-Mtb and the AM80 BMDM experiments and the topically is of great general interest and finding that G-CSF blockade may help block deleterious neutrophil influx is important. However, some of the data provided undercut key conclusions the authors are making in the discussion (and figure legends). These include conclusions around the antimicrobial role of Nox2 in mice and the durability of this CFU difference versus the inflammatory difference. These issues ultimately limit the scope of the paper and limit several of the key conclusions.

Reviewer #3: In the revised version Choi and colleagues have performed additional experiments and re-analysed their data (with ANOVA) to support their conclusion and respond to not only my comments (example: adding new data on bone marrow and lung neutrophils profiling in uninfected KO and WT mice, AMD treatment of infected WT mice), but other reviewer comments as well. Together this new data has improved the manuscript. I am particularly excited to see the data of an in vivo experiment (Fig S5) where authors have infected WT and KO mice by YFP+ Mtb and analysed the uptake of the florescent bacteria by mature and immature neutrophils. They showed (by looking at YFP MFI) the increased permissiveness of immature neutrophils compared to mature one.

**Part II – Major Issues: Key Experiments Required for Acceptance**

Reviewer #1: This reviewer appreciates the inclusion of the experiment using fluorescently labeled bacteria to demonstrate the increased permissiveness in immature neutrophils. Given the caveats introduced by the differences in bacterial burden at this timepoint, additional analysis is required to support the claim that immature neutrophils are more permissive in the setting of NOX2 deficiency (or if there is increased MFI simply because the bacterial burden in the lung is increased). Looking at additional myeloid populations, such as alveolar macrophages and recruited macrophages, and comparing the fold changes in percent infected cells and YFP MFI in these populations as compared to immature neutrophils in WT and KO mice would give insight into these differences. Additionally, this may give some insight into the roles of macrophages during Mtb infection in NOX2 deficiency.

The time course is valuable information to have. However, this time course is in female mice and as the authors state, a major advance of the paper is sex differences in NOX2 deficiency, and the major conclusions of the paper are in male mice. This reviewer realizes that the likely reason this data was shown is that the experiment was already done, but I wonder if the results would be the same in male mice. This is not necessarily required to do, but a mention of this time course being done in female mice and the rationale should be called out in the text, not in a supplemental figure legend where it is hard to find. On a related note, from this time course (albeit in female mice) it appears that week 3 is a very interesting timepoint, since differences in neutrophils are already apparent at a time there is similar CFU between KO and WT mice. This would appear to support the authors’ main conclusion of the paper.

How do the authors reconcile the new results shown of AM80 treatment in WT mice, which had an effect, and anti-G-CSF in WT mice, which did not have an effect? The model proposed is that both primarily mediate their effects through decreasing the numbers of immature neutrophils. If this were true, wouldn’t similar effects from both treatments be expected in WT mice? Was macrophage activation in vivo evaluated during treatment, or just in vitro data? Lastly, the conclusion that T cell responses weren’t affected has the caveat that there are significant differences in bacterial burden at the timepoint evaluated. Couldn’t AM80 have induced an earlier/stronger Th1 response, which is diminished at week 4 due to decreased CFU? Given these caveats, and the strength of the G-CSF results, this reviewer wonders whether this story would be stronger without the inclusion of this experiment.

Reviewer #2: 1. Central to the conclusions of this paper are the CFU differences in male mice shown generally at 4 weeks. At the request of the reviewers a timecourse is now shown that shows what has previously been observed, that the small CFU differences are only seen significantly at 4 weeks post-infection. While the authors say 3 weeks is an increase there are no statistics included and the differences look minor. Importantly, while the authors say the difference in neutrophils goes back to normal, these actually seem to persist with no differences in CFU. In line with other previous studies using Nox2 mice. With these data in the revision, the only supported conclusion is that male Nox2 mice have a transiently higher CFU at 4 weeks, but the paper is written to make these differences in CFU a larger conclusion which is not robustly supported by the data shown.

2. The inhibition of IL1R included in the rebuttal is also confusing. IL1 has a key role in early antimicrobial responses, so blocking IL1R as early as a week following infection is going to have severe effects, as seen by the authors. It is interesting that the male mice succumbed to disease and couldn’t be used in this experiment, but using females complicates comparisons across experiments. Thus, with the way this experiment was done I’m not convinced the data can really discern what the role of IL1a or IL1b is in this model.

Reviewer #3: None

**Part III – Minor Issues: Editorial and Data Presentation Modifications**

Reviewer #1: The authors should entertain additional possibilities (such as other cell types playing a role) as to why the neutrophil depletion did not fully mitigate pathology and decrease bacterial burdens to WT mice. Also, why was the IA8 depletion not efficacious in this setting, when others have depleted pulmonary neutrophils during Mtb infection with much greater effect (DOIs: 10.1038/nmicrobiol.2017.72; 10.1172/JCI130546)?

The methods describing quantification of lung inflammation are still lacking in specifics. How was inflammation determined? Was it with a specific plugin in ImageJ? By eye? If the latter, was the scorer blinded? This information is crucial to ensure reproducibility.

In instances where there is a significant difference between groups, they should not be concluded to be the same. Specific figures to note are: Fig 1E (IL-10 is significantly upregulated in male KO vs WT); Fig S5A-B (the number of total, immature, and mature neutrophils are slightly increased in naïve NOX2 deficient male mice as compared to WT, please change corresponding test); Fig 6D (TNF significantly decreased with AM80 in WT, perhaps choose another word such as “substantially” here?).

Reviewer #2: (No Response)

Reviewer #3: N/A

PLOS authors have the option to publish the peer review history of their article (what does this mean?). If published, this will include your full peer review and any attached files.

Reviewer #1: No

Reviewer #2: No

Reviewer #3: **Yes: **Amit Singhal
---

## [Decision Letter · Decision Letter 2]

12 Aug 2024

Dear Dr. Shin,

We are pleased to inform you that your manuscript 'Permissive lung neutrophils facilitate tuberculosis immunopathogenesis in male phagocyte NADPH oxidase-deficient mice' has been provisionally accepted for publication in PLOS Pathogens.

Best regards,

Christopher M. Sassetti

Academic Editor

PLOS Pathogens

Michael Otto

Section Editor

PLOS Pathogens

Michael Malim

Editor-in-Chief

PLOS Pathogens

orcid.org/0000-0002-7699-2064

Reviewer Comments (if any, and for reference):

Reviewer's Responses to Questions

**Part I - Summary**

Reviewer #1: This study is conceptually exciting, and of interest to the field. I appreciate the detailed responses and careful considerations of the many comments, questions, and criticisms posed by reviewers. I have three relatively minor lingering issues for the authors to address:

1. Language about the role of additional cell types has been included about the AM80 study, yet the title for Figure 6 still states "AM80 administration mitigated TB pathogenesis in male Nox2-/- mice by reducing immature neutrophils." This mechanism of action of AM80 is not supported by the data shown here, and the language should be modified to reflect that this relationship is an association.

2. The image analysis methods are a little hard to follow. Could the authors show example images of processing steps in a supplemental figure?

3. On lines 204 and 206, the authors state there is a discrepancy in CFU values, when Figure 1E shows no significant difference. This language should be modified to reflect that there is no significant difference.

Reviewer #2: The authors have done a good job at addressing my concerns and highlighting possible caveats to their results in the discussion. This is an interesting story that will be appreciated by the TB community.

**Part II – Major Issues: Key Experiments Required for Acceptance**

Reviewer #1: (No Response)

Reviewer #2: (No Response)

**Part III – Minor Issues: Editorial and Data Presentation Modifications**

Reviewer #1: (No Response)

Reviewer #2: (No Response)

PLOS authors have the option to publish the peer review history of their article (what does this mean?). If published, this will include your full peer review and any attached files.

Reviewer #1: No

Reviewer #2: No

---

## [Editor Report · Acceptance letter]

19 Aug 2024

Dear Dr. Shin,

We are delighted to inform you that your manuscript, "Permissive lung neutrophils facilitate tuberculosis immunopathogenesis in male phagocyte NADPH oxidase-deficient mice," has been formally accepted for publication in PLOS Pathogens.

Best regards,

Michael Malim

Editor-in-Chief

PLOS Pathogens

orcid.org/0000-0002-7699-2064